# Tailoring smart hydrogels through manipulation of heterogeneous subdomains

Haoqing Yang[1], Tengxiao Liu[2], Lihua Jin [3], Yu Huang [4] ✉,
Xiangfeng Duan [5] ✉ & Hongtao Sun [1,6] ✉

The mechanical interactions among integrated cellular structures in soft tissues dictate the mechanical behaviors and morphogenetic deformations observed in living organisms. However, replicating these multifaceted attributes in synthetic soft materials remains a challenge. In this work, we develop a smart hydrogel system featuring engineered stiff cellular patterns that induce strain-driven heterogeneous subdomains within the hydrogel film. These subdomains arise from the distinct mechanical responses of the pattern and film domains under applied mechanical forces. Unlike previous studies that incorporate reinforced inclusions into soft matrices to tailor material properties, our method manipulates the localization, integration, and interaction of these subdomain building blocks within the soft film. This enables extensive tuning of both local and global behaviors. Notably, we introduce a subdomain-interface mechanism that allows for the concurrent customization and decoupling of mechanical properties and shape transformations within a single material system—an achievement rarely accomplished with current synthetic soft materials. Additionally, our use of in-situ imaging characterizations, including full-field strain mapping via digital imaging correlation and reciprocal-space patterns through fast Fourier transform analysis of real-space pattern domains, provides rapid real-time monitoring tools to uncover the underlying principles governing tailored multiscale heterogeneities and intricate behaviors.

Soft living organisms, such as human skin and soft invertebrates, often exhibit a remarkable array of complex attributes, including mechanical strength, elastic stretchability, shape transformation, and hierarchical 3D structures[1–4]. The soft tissues in these organisms feature meticulously arranged cellular units (Fig. 1a) that are mechanically intertwined with their neighbors, exerting mutual forces and interacting with their environment. These sophisticated interactions among the integrated cellular unit structures are pivotal in determining the mechanical behaviors and morphogenetic deformations in soft tissues[3,4].

Similarly, cellular grain domains, which serve as the unit building blocks in hard crystalline materials (Fig. 1a)[5], also play an essential role in shaping material properties. The sizes, shapes, arrangements, and responses of these crystal grain domains determine local material characteristics, which in turn dictate the global properties of the material. For example, grain boundaries (GBs), which separate different grain domains, can enhance the mechanical properties of bulk crystals by constraining plastic deformation and impeding dislocation movements between grains[6,7]. Grain size also affects yield strength

[1]The Harold & Inge Marcus Department of Industrial & Manufacturing Engineering, The Pennsylvania State University, University Park, PA, USA. [2]Department of Biomedical Engineering, The Pennsylvania State University, University Park, PA, USA. [3]Department of Mechanical and Aerospace Engineering, University of California, Los Angeles, Los Angeles, CA, USA. [4]Department of Materials Science and Engineering, University of California, Los Angeles, Los Angeles, CA, USA. [5]Department of Chemistry and Biochemistry, University of California, Los Angeles, Los Angeles, CA, USA. [6]Materials Research Institute (MRI), The Pennsylvania State University, University Park, PA, USA. ✉e-mail: yhuang@seas.ucla.edu; xduan@chem.ucla.edu; hongtao.sun@psu.edu

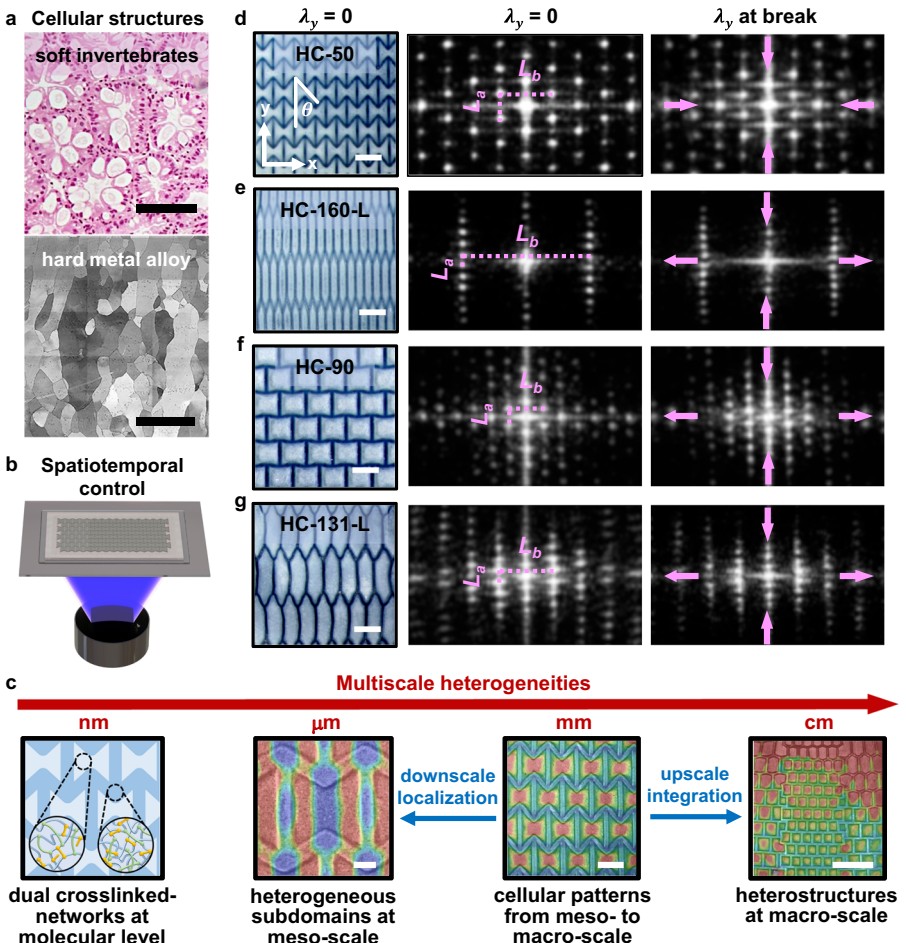

**Fig. 1 | Projection grayscale lithography for spatiotemporal control over soft hydrogel films with cellular pattern domains. a** Illustration depicting cellular structures in soft invertebrates and cellular grain domains in Ti alloys. Histology image reproduced with permission from ref. [1], Springer Nature, and scanning electron microscope image reproduced under the terms of the Creative Commons Attribution 4.0 international license ref. [5], Springer Nature. **b** Projection grayscale lithography via a DLP-3D printer enables spatiotemporal control over the photo-polymerization of hydrogel films for tailored structures and properties. **c** Multiscale heterogeneities in synthetic hydrogels, spanning from molecular-level dual crosslinked networks, meso- to macro-scale cellular units and their localized subdomains, to macro-scale integrations with multiple "phases." **d**–**g** Real-space optical microscopic image of hydrogel films with various pattern domains such as HC-50 ($\theta = 50°$), HC-160-L (longitudinal, $\theta = 160°$), HC-90 ($\theta = 90°$), and HC-131-L (longitudinal, $\theta = 131°$), and their corresponding reciprocal-space patterns through FFT analysis for real-time monitoring of deformations during stretching along the y-direction. The reciprocal-space parameters $L_a$ and $L_b$ correspond to periodic dimension parameters $a$ and $b$ of designed cellular patterns in real space (Supplementary Fig. 3). UV curing times are set to 200 s in cellular pattern domains and 20 s in the film domains; the angle $\theta$ in the shrunk HC-90 hydrogel film is slightly less than the designed 90° due to the shrinkage mismatch between the pattern and film domains (**f**). Scale bars, 500 μm (**a**), 500 μm, 2 mm and 0.5 cm from left to right (**c**), 2 mm (**d**–**g**).

following the Hall-Petch relationship[8]. Additionally, incorporating a second phase can enhance the hardening of metals[9]. These crystallographic features can be tailored through specific material processing, such as transforming a brittle cast structure with nonuniform columnar grains into a wrought structure with finer grains and greatly improved ductility through hot rolling[10].

The characteristics of these cellular unit building blocks, which shape both local and global features, offer invaluable inspiration for developing synthetic soft materials. This is particularly important for applications in various fields such as tissue engineering, biomimetic manufacturing, soft robotics, and artificial muscles[11–20]. Hydrogels, with their high water content and excellent biocompatibility, stand out as a promising candidate in these areas[21–24]. However, achieving sophisticated attributes in synthetic hydrogels remains a formidable challenge. A thorough understanding of the underlying principles is essential for designing smart hydrogel systems with tunable intricate features.

In this work, we develop synthetic smart hydrogels featuring stiff, cellular-like domains to customize their mechanical properties and thermo-responsive deformations. Specifically, by spatiotemporal control of photo-polymerization via dynamic light projection grayscale lithography (Fig. 1b), we print relatively stiff cellular pattern domains that induce strain-driven heterogeneous subdomains within the soft hydrogel film domain under applied mechanical forces. Unlike previous approaches that incorporate rigid inclusions into soft matrices to tailor material properties[25], our method manipulates the localization, integration, and interaction of subdomain building blocks within the soft film, allowing for extensive tuning of both local and global behaviors. This represents a general strategy for designing synthetic hydrogels. Notably, our approach enables the regulation of multiscale heterogeneities, spanning from molecular-level dual crosslinked networks, meso- to macro-scale cellular units with localized heterogeneous subdomains, to macro-scale integrations with multiple "phases," resulting in structural complexities across hierarchical scales (Fig. 1c). Additionally, our work highlights the use of cutting-edge in-situ imaging techniques, including real-space full-field strain mapping via digital image correlation (DIC) and reciprocal-space pattern analysis using fast Fourier transform (FFT) of real-space pattern domains.

These characterizations facilitate a fundamental understanding of the underlying principles governing tailored multiscale heterogeneities and intricate behaviors in these hydrogels.

## Results

### A spatiotemporal control over the photo-polymerization for tunable structures and properties

To fabricate synthetic 2D hydrogel films with a specific thickness, a fixed height projection grayscale lithography setup featuring a set interior spacing (e.g., 500 μm), was built on a Teflon FEP substrate in a digital light processing (DLP)-3D printer. Initially, by temporal control over photo-polymerization, we explored homogeneous hydrogel films to demonstrate their tunable mechanical properties and thermo-responsive deformations. The approach involved using a mix of N-isopropyl-acrylamide (NIPAm) and acrylamide (AAm, 10 mol% of NIPAm) co-monomers, concurrently crosslinked with both a long-chain crosslinker, poly(ethylene glycol) diacrylate (PEGDA, $M_w = 700$), and a short-chain crosslinker, N,N′-methylenebis(acrylamide) (BIS). Our studies indicate that the properties of hydrogels could be systematically tuned by regulating the ultraviolet (UV) light-induced polymerization process. Specifically, the quick gel formation, primarily due to long-chain PEGDA crosslinking of the major monomer NIPAm, allowed for the creation of a hydrogel film within a short exposure period[26,27]. As the curing time increased, the polymer network continued growing, involving more minor monomer AAm and short-chain BIS, leading to copolymer hydrogel films crosslinked with both PEGDA and BIS that showed enhanced ultimate strength and elastic modulus with longer UV exposure times (Supplementary Fig. 1a)[28]. Additionally, the resulting hydrogel is a thermo-responsive smart material, capable of contracting in response to temperatures surpassing the volume phase transition temperature $T_c$. The use of a dual-crosslinker copolymer system facilitates a broad spectrum of shrinking ratios (e.g., $A_{45°C}/A_O = 0.28$-$0.92$) for the printed hydrogel films by varying UV exposure times from 20 to 300 s, thus offering precise control over deformable behaviors (Supplementary Fig. 1b). Moreover, the printed hydrogels exhibit molecular-level heterogeneities, leading to a combination of localized hydrophilicity (contributed by AAm) and hydrophobicity (contributed by NIPAm).

Furthermore, we realized different designs of cellular pattern domains embedded in 2D hydrogel films for tunable features (Fig. 1d–g). Specifically, utilizing a dynamic mask in a DLP-3D printer for spatiotemporal control over photo-polymerization, the relatively stiffer cellular pattern domains (cured for 200 s) were seamlessly integrated into softer hydrogel films (cured for 20 s). For example, one cellular pattern design is a re-entrant honeycomb (HC) metamaterial with a 50-degree angle (HC-50), exhibiting a negative Poisson's ratio of the HC framework ($\nu = -1.485$, Supplementary Fig. 2)[29,30]. The HC-90 design has an HC-structured framework with a 90-degree angle, showing a zero-to-positive Poisson's ratio ($\nu \approx 0 \sim (+0.20)$, Supplementary Fig. 2)[30,31]. Inspired by the behaviors of polycrystals, where the shape of grain domains such as columnar grains influences material properties like anisotropy and ductility[10], we designed two elongated cellular units: HC-131 and HC-160 (Fig. 1e, g). Here, equivalent areas of cellular units are ensured between comparable cellular designs (e.g., $A_{HC-90} = A_{HC-131}$; $A_{HC-50} = A_{HC-160}$). Then, mechanical behaviors for different hydrogel films as varying the angle $\theta$ of the cellular unit were characterized in their shrunk state under a uniaxial tensile test in 45 °C water. Here, FFT analyses of these cellular pattern domains in hydrogel films serve as a real-time monitoring tool to quickly assess their dynamic responses in reciprocal space under tensile tests (Fig. 1d–g). The reciprocal-space parameters $L_a$ and $L_b$ correspond to periodic dimension parameters $a$ and $b$ of designed cellular patterns in real space (Supplementary Fig. 3). For example, in HC-131-L hydrogel films (longitudinal), the parameter $L_a$ in reciprocal space decreased ($\frac{L'_a}{L_a} = 0.814 < 1$, where $L'_a$ represents the parameters in the deformed state, Supplementary Table 1) because of the elongated longitudinal dimension in real space under y-direction stretch, while $L_b$ increased ($\frac{L'_b}{L_b} = 1.104 > 1$, where $L'$ represents the parameters in the deformed state) due to the compressed transverse dimension (Fig. 1g), demonstrating a positive Poisson's ratio. In contrast, the simultaneous reduction of both $L_a(\frac{L'_a}{L_a} = 0.826 < 1)$ and $L_b(\frac{L'_b}{L_b} = 0.940 < 1)$ in reciprocal space for the HC-50 hydrogel (Fig. 1d, Supplementary Table 1) indicated expanding along both x and y directions in real space, a characteristic of a negative Poisson's ratio driven by cellular meta-patterns. The anisotropic mechanical characteristics were further investigated under two stretching directions. When tension was applied along the longitudinal direction, the hydrogel films (e.g., HC-160-L and HC-131-L), featuring columnar cellular units, exhibited considerably higher elastic moduli compared to transverse stretching (e.g., HC-160-T and HC-131-T), underscoring their pronounced anisotropic mechanical properties (Supplementary Fig. 4). In contrast, their counterparts (HC-50 and HC-90) displayed much less variation in elastic modulus when stretched in either direction.

### Strain-engineered heterogeneous subdomains dictating both local and global mechanical properties

Impressively, the patterned HC-50 and HC-90 hydrogel films exhibited simultaneous improvements in ultimate strength, elastic modulus, and material toughness (defined as the energy absorption before rupture, represented by the area under the stress-strain curve) compared to the homogeneous film, indicating enhanced damage tolerance (Fig. 2a). Specifically, these patterned hydrogels showed more than a threefold increase in ultimate strength and an 80–150% increase in material toughness compared to the non-patterned film (Fig. 2b). However, extending the UV curing time in the film domain from 20 to 40 s resulted in an increase in ultimate strength but a decrease in material toughness for both HC-50 and HC-90 hydrogel films relative to the non-patterned counterpart (Supplementary Fig. 5). This trade-off is likely due to a reduced disparity in the elastic modulus between the stiffer pattern domains and the softer film domains[32]. These comparative analyses lead us to hypothesize that the enhanced damage tolerance may be attributed to localized mechanical responses within the soft film domains, influenced by stiff cellular patterns.

To verify this localization phenomenon, we conducted full-field strain mapping of hydrogel films embedded with tracking markers (Supplementary Fig. 6) via in-situ DIC analysis during tensile tests. When stretching the HC-50 hydrogel film along the y-direction (Fig. 2c-f), we observed localized y-strain concentrations in the central film region within each cellular unit ($\varepsilon_y > 45\%$, at a stretch, $\lambda_y = 1.2$) (Fig. 2d). Meanwhile, the x-direction strain map revealed distinct mechanical responses in different local regions (Fig. 2e): a central strip displayed negative x-direction strains ($\varepsilon_x < 0$), while adjacent regions exhibited positive x-direction strains ($\varepsilon_x > 0$), which were confirmed by finite element analysis (FEA) (Supplementary Fig. 7, and detailed parameters for FEA simulation included in Supplementary Table 2). This result indicates that the designed cellular pattern domain induces heterogeneous strain localization within the soft film domain under an applied mechanical load.

We further examined both x- and y-strain profiles along two perpendicular linear pathways (denoted as "X" and "Y") to resolve these heterogeneous regions and their interfaces (Fig. 2d-f). As we stretched the HC-50 film along the y-axis, the increasing y-direction strains underscored disparities in x-direction strains across specific regions within each cellular unit (Supplementary Fig. 8 and Movie 1). These strain localizations help delineate distinct subdomains, such as subdomains A and B (Fig. 2e), which displayed substantial differences in their Poisson's ratios ($\nu_A = +0.10$, $\nu_B = -0.13$, Supplementary Fig. 9a). Notably, the interfaces between these distinct subdomains maintained a static x-strain of around 0%, as indicated by the green-colored boundaries (Fig. 2e) and the x-strain profile along linear pathway X

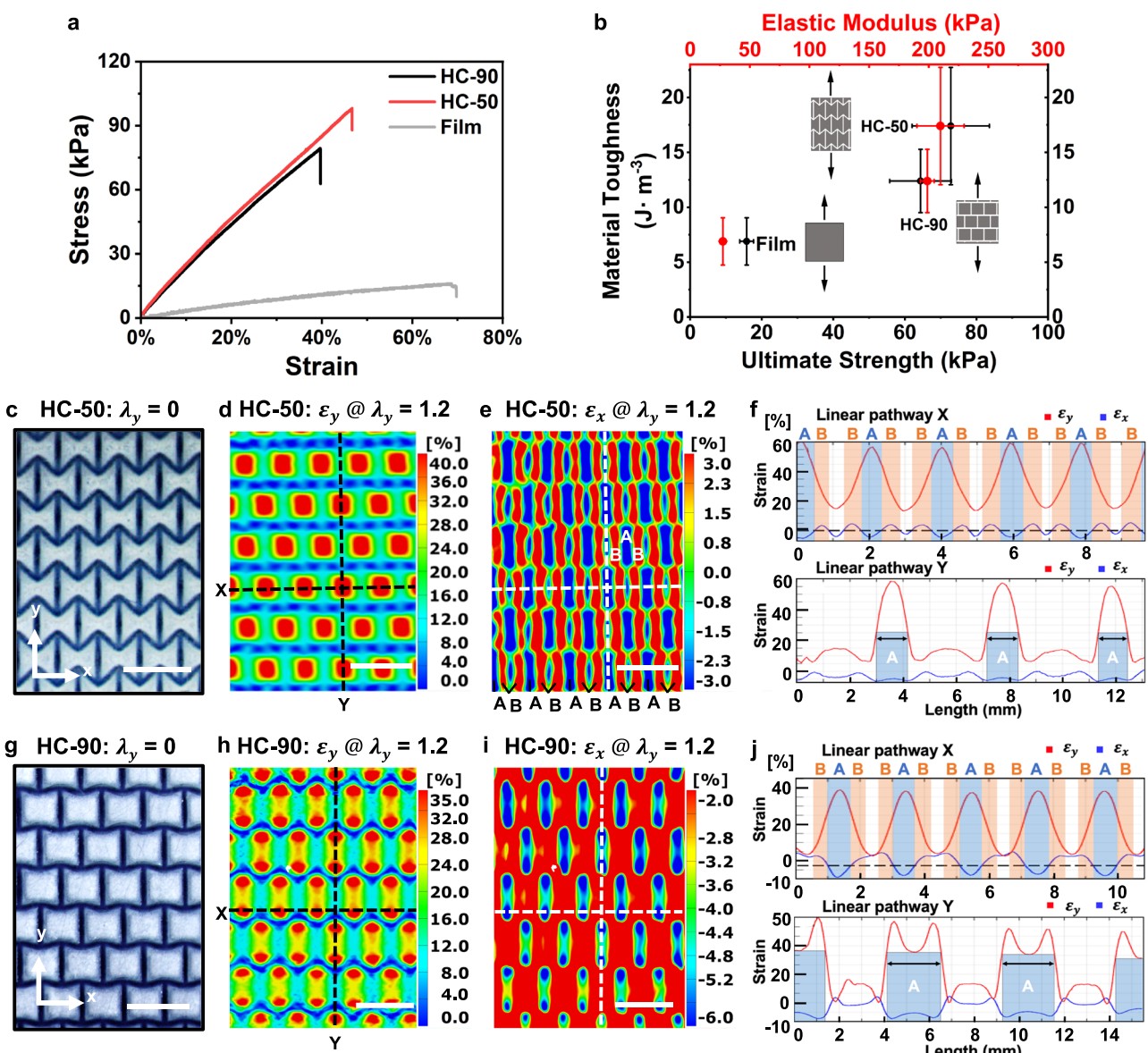

**Fig. 2 | Characterizations of strain-engineered subdomains in dictating both local and global mechanical behaviors. a** Stress-strain curves of 2D hydrogel films with various cellular pattern domains, such as HC-50 ($\theta = 50°$) and HC-90 ($\theta = 90°$), compared to the homogeneous film counterpart. UV curing times are set to 200 s for the cellular pattern domains and 20 s for the film domains and the control film. **b** Correlations between material toughness and ultimate strength/elastic modulus. Data were presented as mean ± SEM, with $n = 4$ replicates. **c–f** Full-field imaging and analysis of HC-50 hydrogel films: optical microscopic image at zero stretch (**c**), strain mappings via DIC analysis for y-strain ($\varepsilon_y$) (**d**) and x-strain ($\varepsilon_x$) (**e**) during uniaxial stretching ($\lambda_y = 1.2$) along the y direction, and strain profiles along two marked linear pathways (**f**). **g–j** Full-field imaging and analysis of HC-90 hydrogel films: optical microscopic image at zero stretch (**g**), strain mappings for $\varepsilon_y$ (**h**) and $\varepsilon_x$ (**i**), and strain profiles along two marked linear pathways (**j**). Scale bars, 3 mm (**c–e**, **g–i**).

(Fig. 2f). The local behaviors of these heterogeneous subdomains, which show uniform mechanical responses within each but pronounced differences across their interfaces, may significantly impact the overall properties of our synthetic hydrogels.

Furthermore, these subdomain building blocks are organized into a supercrystal-like formation, which influences the macroscopic properties. The enhanced damage tolerance observed in our patterned hydrogel films is attributed to the localization and interaction of these heterogeneous subdomains and their supercrystal-like integration, which markedly differs from the conventional strategy of incorporating hard inclusions into soft matrices[25,32]. Under mechanical stretching, subdomains A within the HC-50 film endured substantial local deformation (e.g., $\varepsilon_y > 45\%$ at $\lambda_y = 1.2$, Fig. 2f), enhancing the stretchability and material toughness of the overall film. Subdomains B, characterized by a negative Poisson's ratio, experienced local biaxial

tension, contributing to improved mechanical properties such as elastic modulus and ultimate strength[33]. These less deformed local subdomains B, organized in a supercrystal-like integration, help shield the adjacent, strain-prone subdomain A from further strain amplification, thereby protecting them from potential mechanical failure.

To further investigate the impact of localized heterogeneities on global mechanical properties, we calculated the strain energy density across different local regions, including subdomains A and B within the ductile film domain and the stiff pattern domains (see details in Supplementary Note, Supplementary Fig. 10, and Supplementary Tables 3–6). At a stretch of 1.2 (20% global strain) within the elastic deformation region, subdomains A and B in the HC-50 sample experienced larger local deformations ($\varepsilon_A = 45 - 57\%$; $\varepsilon_B = 17 - 41\%$) compared to the homogeneous thin film ($\varepsilon_{film-20s} = 20\%$) cured for the same duration (20 s) (Supplementary Table 3). Consequently,

calculated strain energy densities of these heterogeneous subdomains were $u_A = 5.98\,\mathrm{J\,m^{-3}}$ and $u_B = 3.34\,\mathrm{J\,m^{-3}}$, which are five to ten times those of the homogeneous thin film under the same global strain of 20% ($u_{film-20s} = 0.60\,\mathrm{J\,m^{-3}}$) (Supplementary Table 3). These elevated strain energy densities in the local subdomains contribute to the increased material toughness of the overall patterned hydrogels.

In addition to reinforcement from the stiff pattern domains (elastic modulus, $E_{200s} = 635\,kPa$), the high strain energy in subdomains B, combined with the biaxial strain caused by the negative Poisson's ratio effect of the meta-structured patterns, offered enhanced resistance to deformation and stress. This led to improvements in both the global elastic modulus and ultimate strength, consistent with previous studies showing that auxetic inclusions can enhance the overall composite's mechanical properties[33].

By altering the angle $\theta$ of the cellular unit from $50° – 90°$, the HC-90 hydrogel film revealed the emergence of "twin" sites within subdomain A, as identified by strain mapping and strain profiles along the linear pathway Y (Fig. 2h–j, Supplementary Figs. 11, 12, and Movie 2). Like the HC-50 design, the HC-90 variant also exhibited noticeable differences in Poisson's ratios between adjacent subdomains (Supplementary Fig. 9b). Moreover, FEA simulation results illustrated the formation of dumbbell-shaped subdomain A by further extending the angle $\theta$ to 120° (e.g., HC-120, Supplementary Fig. 13). Therefore, the distinct responses of these localized strain-induced subdomain building blocks, governed by the stiff pattern domains, enable extensive tuning of both local and global behaviors (Fig. 2).

However, increasing the local UV curing time in the film domain reduced the localization effects on global mechanical features. Specifically, as UV exposure in the soft film regions increased from 20 to 40 s, the strain energy density in subdomains A of the HC-50 hydrogels decreased from 5.98 J m⁻³ (20 s) to 3.79 J m⁻³ (40 s) at a stretch of 1.2 (Supplementary Tables 3 and 5), indicating reduced discernibility of strain-induced heterogeneities within the soft domains (Supplementary Figs. 5, 14). Meanwhile, a higher strain energy density was observed in the vertical regions of the stiff patterns (3.84 J m⁻³ for 20 s vs. 6.22 J m⁻³ for 40 s, Supplementary Tables 3 and 5) as UV exposure in the soft film domains increased, suggesting that the stiff patterns increasingly bore the load rather than inducing localized heterogeneous strain in the soft film domains. Therefore, a substantial mismatch in elastic modulus between the pattern and film domains is crucial for inducing localized heterogeneous subdomains, which leads to a synergy between strength and toughness. These findings highlight the effectiveness of cellular pattern designs in manipulating local subdomains and their collective attributes at the macroscopic level, offering a general approach to developing synthetic soft materials with tailored mechanical properties.

## Integration of various "phases" for manipulating mechanical features

In addition to the downscale localization (e.g., heterogeneous subdomains), we further explore the upscale integration of various cellular patterns (representing different "phases") within a unified hydrogel film. For instance, we integrated two distinct cellular patterns (HC-50 and HC-160-L) into a unified hydrogel film, thus creating an incorporated HC-160-L&HC-50 configuration. Real-time strain mapping (Fig. 3a and Supplementary Fig. 15) revealed areas of y-strain concentration within the more elastic "HC-50 phase" (e.g., $\varepsilon_{y_{P1}} = 93\%$ after a 150-s stretch, Supplementary Fig. 16). In contrast, the stiffer "HC-160-L phase" (e.g., $\varepsilon_{y_{P2}} = 8\%$, Supplementary Fig. 16) contributed to an increased elastic modulus of the integrated HC-160-L&HC-50 "multi-phase" hydrogel, albeit with reduced stretchability compared to an HC-50 "single phase" hydrogel (Supplementary Fig. 16). Moreover, two sets of time-resolved FFT patterns demonstrated different responses in reciprocal space during stretching, tracking distinct

deformations between the stiffer "HC-160-L phase" (marked by yellow ovals) and the softer "HC-50 phase" (highlighted by pink circles) (Fig. 3b).

We also embedded a stiffer "HC-160 phase" inclusion into a more elastic "HC-50 phase" matrix within our synthetic hydrogels. Various arrangements of these inclusion "phase", including longitudinal (HC-160-L) and transverse (HC-160-T) orientations, were compared to examine the coherence levels between the inclusion and the surrounding matrix (Fig. 3c-f, Supplementary Fig. 15). Aligning the longitudinal direction of the elongated cellular units in stiffer HC-160-L inclusion with the stretching direction increased the local resistance to deformation (e.g., $\varepsilon_{y_{P3}} = 47\%$, Supplementary Fig. 16b) in the associated matrix area, as indicated in the solid rectangle region (Fig. 3c, Supplementary Fig. 15). This contrasts with the dashed rectangle region, which was less influenced by the inclusion (e.g., $\varepsilon_{y_{P4}} = 93\%$, Supplementary Fig. 16b). In contrast, hydrogels containing transversely oriented "HC-160-T phase" displayed barely improved elastic modulus of the overall sample (Supplementary Fig. 16a). Additionally, regions near the interface of the HC-160-T inclusion (e.g., P6 in dashed rectangle region, Fig. 3e and Supplementary Fig. 15) demonstrated a negative Poisson's ratio ($\nu_{P6} = -0.22$, Supplementary Fig. 16c), whereas the central region of the inclusion (e.g., P5 in solid rectangle region) exhibited a positive Poisson's ratio ($\nu_{P5} = +0.15$). Thus, the strategic integration and manipulation of multiple "phases" and their interfaces enable fine-tuning of both local and global mechanical properties. The selective local straining in these integrated structures suggests potential application as substrates for stretchable electronics, where the stiffer phases support relatively rigid electronic components.

Additionally, the dimensions of subdomains and their interfacial boundaries play crucial roles in dictating the hydrogels' mechanical behaviors. To investigate the size effect, we fabricated a series of hydrogel films with varying critical dimensions of cellular units (Fig. 3g and Supplementary Fig. 17), which illustrated supercrystal-like configurations of heterogeneous subdomains across all samples. We controlled the thickness t of the cellular walls in proportion to the size of the cellular unit, ensuring the same area fractions of the stiffer pattern domains among various samples. In both HC-50 and HC-90 variants, the sizes of the subdomains consistently decreased with the reduction in dimensions of cellular units under mechanical load (Supplementary Table 7). This, in turn, noticeably increased the number of subdomains and their interfacial boundaries within the same footprint area.

We studied the size-dependent mechanical properties of these hydrogels to elucidate a subdomain-interface mechanism. Stress-strain curves highlighted that a decrease in the sizes of cellular units and their associated subdomains led to simultaneous increases in elastic modulus and ultimate strength (Fig. 3h, i). Furthermore, we analyzed the relationship between ultimate strength and critical dimensions across a series of patterned hydrogel films with varying cellular units (e.g., $d_A$, $d_B$, and $d$ in Fig. 3j and Supplementary Fig. 18). With smaller subdomains and more abundant interfacial boundaries, the enhanced interface effect more effectively mediated the contrasting local characteristics, thereby enhancing the global mechanical properties of our designed hydrogels. Building on the demonstrated subdomain-interface mechanism, we integrated two distinct "phases" with differently sized cellular units (e.g., $d = 0.99$ and 2.13 mm) (Supplementary Fig. 19). Under tensile stretching, real-space strain mappings and reciprocal-space FFT patterns exhibited distinct local responses from these two different "phases". Specifically, the phase with smaller cellular units showed less local strain (P2 and P4) during stretching, indicating relatively stiffer characteristics, in contrast to the phase with larger cellular units (P1 and P3), which underwent greater local strains.

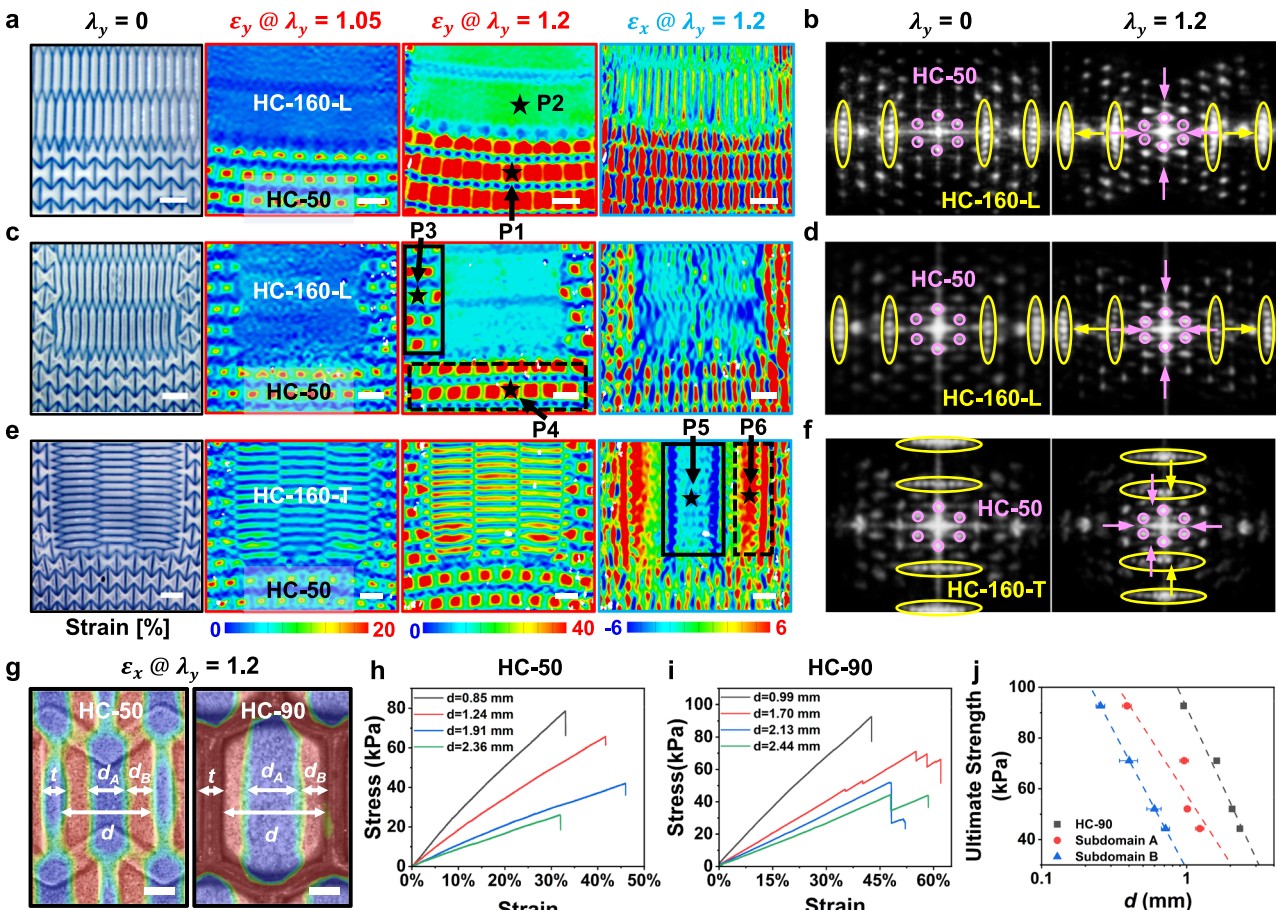

**Fig. 3 | Mechanical characterizations of synthetic hydrogel material system with various designs. a, b** Illustration of "multi-phase" structure as depicted in HC-160-L&HC-50 hydrogels through full-field strain mapping in real space (**a**) and dynamic FFT patterns in reciprocal space (**b**), with stretching applied along the y-direction. The UV curing times are set at 200 s for cellular patterns and 20 s for the film. **c–f** Incorporation of inclusion "phases" into synthetic hydrogels, including HC-160-L@HC-50 (**c** and **d**) and HC-160-T@HC-50 (**e** and **f**), characterized by strain mapping (**c** and **e**) and dynamic FFT patterns (**d** and **f**) during stretching along the y-direction. The HC-160 and HC-50 "phases" are denoted by yellow ovals and pink circles, respectively, in the reciprocal space (**b–f**). **g** Defined critical dimension parameters of cellular units and subdomains in HC-50 and HC-90 hydrogel films at a y-direction stretch ($\lambda_y$) of 1.2. $d_A$ and $d_B$ represent the width of subdomains A and

B, respectively; $d$ is calculated as ($d_A + 2 \times d_B$); t denotes cellular wall thickness. The full-field strain mapping results are displayed as superimposed images, integrating optical grayscale images of the hydrogel samples with strain maps to simultaneously capture both structure and deformation information. **h, i** Stress-strain curves for HC-50 (**h**) and HC-90 (**i**) hydrogel films with cellular patterns of varied sizes under stretching along the y-direction. **j,** Established relationships between ultimate strength and critical dimensions (e.g., $d$, $d_A$, and $d_B$) for HC-90 hydrogel films. Data were presented as mean ± SD, with $n$ = 3 replicates. P1, P3, and P4 refer to the subdomain A of the "HC-50 phase", P2 refers to the film domain of "HC-160-L phase", P5 and P6 refer to the central region and near the interface within the "HC-160-T inclusion phase". Scale bars, 2 mm (**a, c, e**), 500 μm (**g**).

## Co-designing shape transformation and mechanical properties

Mechanical and deformable behaviors represent two important characteristics of smart soft materials. By regulating relatively stiffer cellular patterns as thermo-responsive actuation backbones, we can control the shrinkage of local softer film domains within each cellular unit, thereby manipulating the global deformation of the entire 2D hydrogel film. For instance, by tuning the local curing time of cellular patterns from 40 to 200 s, we achieved a wide range of global shrinking ratios from 0.40 to 0.85 across various HC-90 hydrogel films (Supplementary Fig. 20). To emulate the complex interplay between deformable and mechanical behaviors observed in soft living organisms, we devised a co-design strategy enabling simultaneous adjustment of both aspects within a single synthetic hydrogel.

We commenced by investigating the correlation between mechanical properties and shrinkage-induced deformations in HC-90 hydrogel films with varying cellular unit sizes (e.g., $d$ = 0.99–2.44 mm, Fig. 4a) while spatially tuning the local curing time of the cellular patterns. Increasing the curing time revealed a trade-off between enhanced elastic modulus and large shrinkage (lower shrinking ratio $A_{45°C}/A_O$) in hydrogel films with identical cellular unit sizes. However,

altering the size of cellular units offered extensive tunability of the elastic modulus while maintaining similar shrinkage levels. For example, within a narrow range of shrinking ratios ($A_{45°C}/A_O$ = 0.78-0.81), highlighted by a vertical red band (Fig. 4a), an HC-90 hydrogel film with smaller cellular units ($d$ = 0.99 mm) exhibited an elastic modulus of 176 KPa, more than twice that of the HC-90 film (e.g., 80 KPa) with larger cellular units ($d$ = 2.13 mm). These variations in mechanical properties are primarily due to the subdomain-interface mechanism discussed earlier. Thus, by co-designing cellular pattern domains with consideration of both shrinkage and size effects, we can decouple the control over deformable and mechanical features within a single hydrogel material.

To demonstrate our co-design approach, we introduced shrinkage-induced gradient and "multi-phase" heterogeneities into a single hydrogel film. By varying the curing times of local cellular patterns from 60 s at the top to 200 s at the bottom (Fig. 4b), we established a shrinking gradient within the thermo-responsive cellular backbones. This gradient controls the variation in local shrinkage across the cellular units, resulting in a global shrinking gradient throughout the entire hydrogel film. Concurrently, the integrated

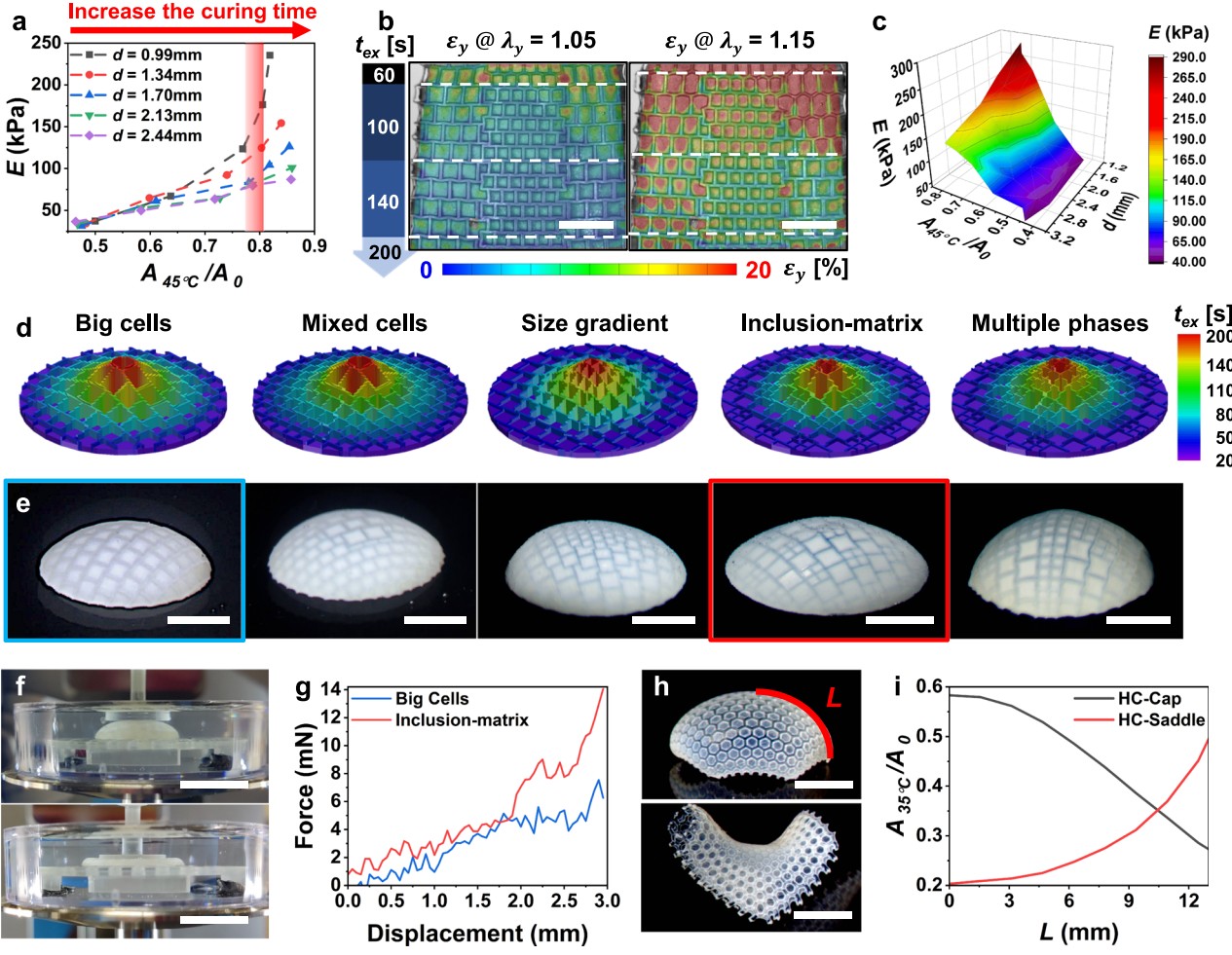

**Fig. 4 | Co-designing shape transformations and mechanical properties through the simultaneous encoding of gradients and "multi-phase" heterogeneities. a** Correlations between elastic moduli and shrinkage ratios for HC-90 hydrogels with varying cellular unit sizes ($d = 0.99$–$2.44$ mm) under different local curing times of cellular patterns ranging from 40 to 200 s. **b** Strain mapping of a hydrogel film with simultaneously encoded gradients (achieved by varying local curing times from 60 to 200 s in the cellular pattern domains, with a consistent 20-s curing time in the film domains) and an integrated stiffer "phase" under progressive stretching. Full-field strain mapping results are displayed as superimposed images, integrating optical grayscale images of the hydrogel samples with strain maps to capture both structure and deformation information simultaneously. **c** A

3D master surface displaying the correlations among shrinkage ratios, elastic moduli, and cellular unit sizes. **d, e** Designs of non-Euclidean caps featuring different arrangements of heterogeneous "phases": spatial distribution of curing times (**d**) and the resulting 3D caps after shape transformation (**e**).
**f, g** Compression tests conducted in 45 °C water on representative 3D caps, showing the axial load application (**f**) and force-displacement correlations (**g**).
**h, i** Encoding smooth gradients for precise control of 3D shapes (**h**), along with the corresponding growth functions ($\eta = A_{35°C}/A_0$) dictating the shape transformations (**i**). The variable arc length on the surface is represented by $L$ in (**i**). NIPAm monomer was selected over co-monomers to lower the $T_c$ of the hydrogels (**h** and **i**). Scale bars, 5 mm in (**b, e, h**), 1 cm in (**f**).

stiffer "phase," composed of smaller cellular units, experiences less strain compared to the relatively softer "phase," which consists of larger cellular units, leading to mechanical reinforcement (Fig. 4b).

Furthermore, we have developed a 3D master surface enabling the programming of 2D-to-3D shape transformations and tailored mechanical features (Fig. 4c), providing multifaceted controls compared to the simpler 2D master curve reported in previous studies[26,34,35]. Our 3D model unravels the interplay among shrinkage, elastic modulus, and cellular unit sizes, providing a possible solution for simultaneously directing both shrinkage-induced deformations and mechanical properties within a single hydrogel material system.

For example, we programmed five 2D hydrogel films to undergo shape transformations into non-Euclidean 3D caps, each following different design pathways, such as varying combinations of cellular sizes on the 3D master surface. To spatiotemporally encode 2D films, we designed a gradient using concentric circles, with rings of low-shrinking domains (under high exposure times) encircled by high-shrinking domains (under low exposure times), as evidenced by the

spatial distribution of exposure times (Fig. 4d). These programmed in-plane dimensional changes facilitate the conversion of 2D films into prescribed 3D structures via out-of-plane buckling, resulting in loosely retained similar 3D cap shapes with Gaussian curvatures>0 (Fig. 4e)[26,35]. The resulting 3D shapes are formed through a sequential 2D-to-3D shape transformation within the temperature range of 35 °C to 45 °C. More significantly, these shape-morphed 3D caps, integrated with different arrangements of multiple "phases," replicate mechanical enhancements demonstrated in their 2D counterparts (Fig. 3). Compression tests, applying load along the axial direction, demonstrated that the 3D cap containing stiffer inclusions withstood larger forces for a given displacement compared to the cap without inclusions (Fig. 4f, g). This work provides a possible solution for simultaneously customizing and decoupling shape transformations and mechanical enhancements within a single material system— an achievement rarely accomplished in prior studies on synthetic smart materials[16,36].

Aiming for potential biological applications, we opted for the NIPAm monomer instead of co-monomers to lower the $T_c$ of our

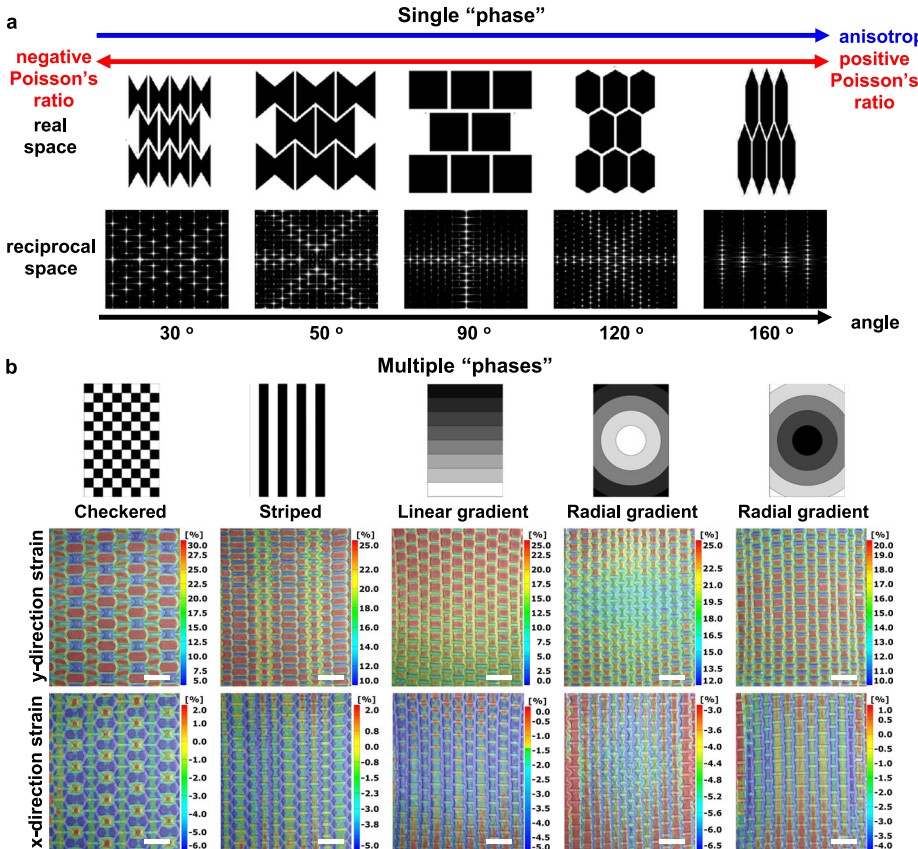

**Fig. 5 | A library of different single "phase" designs and their intricate integrations of multiple "phases". a** Alteration of the angle of cellular backbones in a "single phase" hydrogel, indicating extensively adjustable material properties. **b** Incorporation of diverse "phases" into five distinct "multi-phase" configurations characterized by full-field strain mapping. Strain mapping results are displayed as superimposed images, integrating optical grayscale images of the hydrogel samples with strain maps to simultaneously capture both structure and deformation information. Scale bars, 3 mm (**b**).

hydrogel, enabling shape transformations near physiological temperatures (e.g., 35 °C, Supplementary Fig. 21). In these experiments, hexagon-shaped cellular patterns (HC-120, $\theta = 120°$) were employed to encode smooth gradients for the shape transformations (Fig. 4h, i). Specifically, growth functions η were used to precisely program the gradient with concentric hexagon circles (Supplementary Fig. 22). This approach allows for fine-tuning curvatures and angles in the shape-morphed 3D configurations, including caps (Gaussian curvature > 0) and hyperbolic saddles (Gaussian curvature < 0) (Supplementary Fig. 23)[15,26].

## A library of various single "phase" designs and their "multiple phase" integrations

To further explore a universally applicable method for creating synthetic hydrogels with customizable characteristics, we have elaborated a library of various cellular units to manipulate material features. By varying the angle θ of these cellular units within a "single-phase" hydrogel, we achieved extensive tunability in properties, including anisotropic characteristics and a range from negative to positive Poisson's ratios (Fig. 5a). Additionally, we have compiled "multi-phase" hydrogels by strategically integrating different cellular units, allowing for more complex features (Fig. 5b). Specifically, we demonstrated five configurations[37], each exhibiting distinct heterogeneous properties under tensile load.

For example, configurations such as checkered and striped patterns amalgamate phases with opposing Poisson's ratios (e.g., negative Poisson's ratio: HC-50, $\theta = 50°$; positive Poisson's ratio: HC-130, $\theta = 130°$), fostering local mechanical diversity. Notably, the "HC-50 phase" exhibiting a negative Poisson's ratio showed much lower local

y-strain compared to the "HC-130 phase" with a positive Poisson's ratio. This disparity arises from the localized biaxial tension behaviors led by the HC-50 meta-structured cellular backbone, enhancing resistance to deformation. Additionally, we introduced graded configurations like linear and radial gradients, where the angles θ of cellular patterns undergo progressive changes. In the linear gradient, the angle transitions from 50° (bottom) to 130° (top), resulting in noticeable subdomain evolution and a shift in local Poisson's ratios from negative to positive (Fig. 5b). In radial gradient structures, the angle in cellular units increases or decreases radially from the central region to the periphery, generating controlled strain gradients observable in both x- and y-directions under tensile stress. These "multi-phase" configurations, which allow for the manipulation of specific local behavioral characteristics, show great potential for applications in soft robots and wearable devices. By enabling different local responses under the same external force, they facilitate the handling of complex objects and accommodate the anatomical intricacies of moving organisms.

## Discussion

In summary, we have introduced a general strain-engineering approach for designing synthetic smart hydrogels with customizable local and global features. Our study reveals five main findings: (1) spatiotemporal control over photo-polymerization in our smart hydrogel system, enabled by advanced projection grayscale lithography, allows comprehensive regulations of material properties and hierarchical structures; (2) the manipulation of the interplay among localization, integration, and interaction of strain-engineered heterogeneous subdomains substantially influences both local and global characteristics of the synthetic hydrogels under external mechanical

forces; (3) the successful application of engineered cellular units and their associated downscale subdomains and upscale integration to our synthetic hydrogels offers an effective strategy for designing soft materials at hierarchical scales, diverging from the conventional molecular-level design of polymer materials; (4) our use of in-situ imaging techniques (e.g., DIC, FFT) marks the instance of visualizing strain-induced heterogeneous subdomains and their sophisticated organization and interactions in soft smart materials, offering rapid, real-time monitoring tools for exploring multiscale heterogeneities and intricate dynamic behaviors; and (5) a pioneering subdomain-interface mechanism enables a co-design strategy that simultaneously customizes and decouples multifaceted features (e.g., mechanical properties, shape transformations, and hierarchical 3D structures) within a single synthetic smart material system, which can rarely be achieved in current synthetic soft materials. Thus, our work presents a promising solution for developing synthetic smart materials with sophisticated characteristics akin to those observed in natural living systems. This approach holds particular potential for applications in fields such as tissue engineering, soft robotics, artificial muscles, soft exoskeletons, and wearable devices.

## Methods

### Materials

High-purity chemicals were purchased from the following sources: N-Isopropylacrylamide (NIPAm, ≥97%), N, N'-Methylenebis(acrylamide) (BIS, ≥97%), and Phenylbis(2,4,6-trimethylbenzoyl) phosphine oxide (PBPO, ≥97%) from Fisher Scientific; Poly(ethylene glycol) Mn 700 (PEGDA 700) and acrylamide (AAm, ≥99%) from Sigma-Aldrich; Ethanol 200 proof (100%) and Acetone (≥99.5%) from VWR; and Amine Functionalized Graphene Oxide (A-GO, -15 μm) from MSE supplies (Arizona, US). All chemicals were used as received without further purification.

### Preparation of printing resins

Photosensitive resins were formulated using NIPAm and AAm as co-monomers. Specifically, NIPAm (7 g), AAm (10 mol% of NIPAm), PEGDA (long-chain crosslinker, 1.1 mol% of NIPAm), BIS (short-chain crosslinker, 2.2 mol% of NIPAm), and PBPO (photoinitiator, 0.765 mol% of NIPAm) were dissolved in 10 mL of ethanol. For NIPAm monomer-based resins, NIPAm (4 g), PEGDA (1.25 mol% of NIPAm), BIS (1 mol% of NIPAm), and PBPO (0.15 mol% of NIPAm) were mixed in a 10 mL aqueous solution (water to acetone ratio: 1:3 ratio by volume). The mixtures were then stirred in amber glass bottles using a magnetic stirrer until homogeneous. For DIC analysis, A-GO (0.6 mg mL$^{-1}$) was added to the NIPAm-AAm ethanol solution and ultrasonicated for 0.5 h. All resins were ultrasonicated for 15 min immediately before use.

### Projection grayscale lithography of hydrogel films

Digital light projection grayscale lithography was performed using an Anycubic Photon D2 digital light processing (DLP)-3D printer (intensity and wavelength of the ultraviolet light: 2.5 mW cm$^{-2}$, 405 nm). A polydimethylsiloxane (PDMS) spacer with a thickness of 500 μm was placed on a Teflon FEP film to build a projection lithography cell. The resin was injected into the cell, covered with a glass coverslip, and exposed to digital light for polymerization. The designs of homogeneous films and engineered films with cellular-patterned domains were conducted using CAD SolidWorks, with the thickness of the 3D model determining localized exposure times $t_{ex}$ ($t_{ex} = \frac{model\,thickness}{layer\,thickness\,of\,the\,slices} \times normal$ exposure times per layer). Stereolithography (STL) files were processed into 2D slices with local exposure time information (Supplementary Fig. 22). Post-polymerization, hydrogel films were detached from the glass coverslip, immersed in ethanol for 1 min to remove unreacted monomers, crosslinkers, and photoinitiators, and then hydrated in DI water at 4 °C for 12 h, with water change every 6 h.

### Measurement of areal swelling and shrinking ratios

Homogeneous films (5 mm × 5 mm) underwent uniform UV exposure for 8–300 s to study the swelling and shrinking ratios of the hydrogels (Supplementary Fig. 1). Swelling measurements involved placing the printed samples in DI water at 25 °C for 6 h until stable. The swelled state area ($A_{25°C}$) was measured, and areal swelling ratios were calculated as $A_{25°C}/A_0$, where $A_0$ is the area of as-printed hydrogel film. For shrinking measurements, NIPAm-AAm samples were first hydrated at 25 °C for 2 h, then gradually heated to 45 °C (5 °C per hour). A similar process was applied to NIPAm samples, heating to 35 °C (0.5 °C per hour). Surface area measurements were taken using a Trinocular Stand Stereo Zoom Microscope (7X-45X, AmScope) and analyzed by ImageJ (NIH, USA).

### Mechanical tests of synthetic hydrogels

Tensile tests for 2D hydrogel films were performed on a UniVert mechanical tester (CellScale, Canada). To maintain the shrunk state, the water tank was preset to 45 °C. Tensile tests involved a 20 Hz, 20 mm gauge length, and 1.2 mm/min stretching rate. Sample thickness was measured with a digital caliper. Elastic modulus was determined from the initial slope of the stress-strain curve (1–5% strain range), and tensile material toughness was calculated from the area underneath the stress-strain curve. Compression tests were conducted on 3D shape-morphed caps using a linear drive in a compression mode of a dynamic mechanical analyzer (DMA, Anton Paar MCR 702e). A preload of 10 mN was applied between the two platens to the 3D cap, which was then compressed at a rate of 0.1 mm·s$^{-1}$ underwater at a temperature of 45 °C for the test. Additionally, a compression test conducted without a sample was performed to obtain the buoyancy data of the platen, enabling precise force calculations on the 3D cap. The non-zero starting point of the force data is due to slight vibrations in the buoyancy at the beginning.

### Full-field strain mapping via DIC

In-situ DIC analyses were conducted using GOM Correlate software (Trilion, PA, USA). A 12 MP 2D sensor with a minimum spatial resolution of 25 μm (refers to the minimum area over which the strain can be calculated) was positioned in front of the hydrogel samples to capture full-field surface images during tensile tests for DIC analysis. The image of the undeformed sample served as a reference. To better evaluate heterogeneous mechanical responses via strain mappings, Amine Functionalized Graphene Oxide (A-GO - 15 μm) powders, acting as markers, were mixed with the resin (0.6 mg mL$^{-1}$): the measured marker size (-20 μm) <point distance between facets (-30 μm) <measured minimum structural dimensions (-100 μm). The DIC capture frequency was set at 2 Hz. Real-time, full-field displacements and strains were derived from time-resolved 2D grayscale images, enabling detailed dynamic mechanical analysis at specific sites. The full-field strain mapping results were presented in two ways: (1) as superimposed images combining the optical grayscale images of the hydrogel samples with strain mapping to simultaneously show both structural and deformation information (e.g., Figs. 1c, 3g, 4b, 5b, Supplementary Figs. 5, 8, 12, 15, 17, 19), or (2) as colorful strain mapping images without overlapping the optical image of the sample to better visualize the heterogeneous subdomains (e.g., Fig. 2d, e, h, i, and Fig. 3a, c, e).

The local strains ($\varepsilon_x$ and $\varepsilon_y$) and their time-dependent variations for any specific sites were derived from time-resolved full-field strain maps. Poisson's ratios of subdomains were calculated using the formula $\nu = -\frac{d\varepsilon_x}{d\varepsilon_y}$. This involved selecting four key tracking points at the left, right, bottom, and top. These points allowed us to measure deformations along the x and y axes, calculated as $\triangle x = x_{right} - x_{left}$ and $\triangle y = y_{top} - y_{bottom}$, respectively (Supplementary Fig. 9). Subdomains A and B were determined by analyzing the selective strain profiles along two perpendicular linear pathways intersecting the subdomains (Fig. 2f, j, Supplementary Figs. 8, 12, and Supplementary

Movies 1, 2). The boundary regions, characterized by minimal increases in local x-direction strain during stretching, were identified as the interfaces between heterogeneous subdomains.

## FFT analysis

The movements of intersection node points in cellular patterned domains were tracked in real-space images and correlated with their FFT patterns in reciprocal space during tensile testing. Specifically, real-space images generated by GOM Correlate software were converted to binary format and processed through FFT in ImageJ software, providing a dynamic and comprehensive view of the deformations under mechanical load.

## Finite element analysis

FEA was conducted through simulations in SolidWorks software to validate the mechanical features of our synthetic hydrogels, which were obtained from the in-situ DIC analysis during tensile testing. The elastic modulus values for various hydrogel materials, cured for durations such as 20, 40, and 200 s, were derived from experimental tensile tests, which were used in the FEA simulations. Key parameters used in FEA simulation were summarized in Supplementary Table 2.

## Data availability

The data supporting this study's findings are available in the Article and its Supplementary Information. Source data are provided with this paper.

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

## Acknowledgements

H.S. acknowledges financial support from the Pennsylvania State University start-up fund.

## Author contributions

H.S. conceived the research. H.Y conducted the computer-aided design, fabrication, mechanical tests, and in-situ imaging characterizations. T.L. and L.J. contributed to data interpretations and post analyses. H.S., X.D., and Y.H. supervised the research. H.S., X.D., Y.H., T.L., and H.Y. co-wrote the manuscript with input from all the authors. All authors discussed the results and commented on the manuscript.

## Competing interests

H.S. and H.Y. are inventors on patents (US Provisional Application No. 63/695,004) relating to this study filed by the Pennsylvania State University, University Park. H.S. and H.Y. have no other competing interests. The remaining authors declare no competing interests.
