## [Transparent Peer Review file · Nature Communications]

Tailoring Smart Hydrogels Through Manipulation of Heterogeneous Subdomains

Corresponding Author: Professor Hongtao Sun

Version 0:

Reviewer comments:

Reviewer #1

(Remarks to the Author)

In this manuscript, the authors developed a kind of smart hydrogels by forming strain-engineered heterogeneous subdomains within a single material system. In-situ imaging characterization was used to explore the multiscale heterogeneities and explain the intricate behaviors. It is very impressive to see both negative and positive x-direction strains. The contents are well organized. Although this work is of good innovation, there are some problems to be addressed by appropriate revisions. I recommend an acceptance of this manuscript after minor revisions. Other comments and issues are listed below.

1. In the introduction part, the authors mentioned the complexity of biological tissues and the crystallographic features of inorganic metals, but the gels prepared in this work are structurally mimicking the metals. The relationship between the gels and the biological tissues needs to be clarified.

2. It is very good to use in-situ imaging to characterize the structures under loadings. The conclusions obtained from this characterization should be explained more clearly. In addition, more detailed experiment procedures should be described.

3. The gel materials are composed of hard skeleton and soft domains. It seems that the main properties of this kind of materials are contributed by the hard skeleton. What is the role of soft domains? How about the properties of the structure with only the hard skeleton?

4. In page 5, the authors said that PEGDA will crosslink the polymer first and then the BIS. Why the crosslinking does not happen at the same time?

5. In Figure 3j, the authors claimed that ultimate strength and the inverse square root of the critical dimensions showed linear relationship. Is there any physical meaning of this scale relationship? For better visualization, a semi-log plot should be more appropriate.

Reviewer #2

(Remarks to the Author)

This paper is based on the clever idea patterning a hydrogel into a cellular pattern of stiffer and less stiff regions. It is somewhat analogous to the extensive work on 3D printed meta-materials except that the effect results from differential crosslinking and instead of the open space in 3D printed materials the spaces between the stiff cell walls are filled with a lower modulus material. The resulting materials exhibit customizable mechanical properties, enhanced strength and toughness, and thermo-responsive deformations.

The authors return several times to an analogy to metal polycrystalline materials. This analogy, in my opinion, overstated and not necessary to the point of the paper. For example the word "precipitate" is used though there are in fact zero precipitates in the systems studied. There is an analogy to the Hall-Petch relation that has no physical basis. I would recommend eliminating this analogy – it would make the paper easier to read and not distract from the really interesting elements of this work. Along similar lines the analogies to biological materials don't really add much to the paper. How

would the structures designed here resemble actual hierarchical structure of biomaterials?

On the title of the paper: I don't see anywhere that strain induces heterogenous subdomains. Rather the subdomains are baked in by the patterning due to differential polymerization. In my opinion the title needs revision, perhaps to something like "Tailoring Smart Hydrogels Through Manipulation of Subdomains of Local Stiffness."

An additional recommendation is that the authors do more to strengthen the mechanical aspects of the study, particularly related to the mechanics underlying the observed stiffening and toughening effects in the patterned hydrogel composite structures. While these effects are evident from uniaxial tension tests and anticipated based on the material properties of the subdomains compared to the matrix, it is not immediately clear how such local heterogeneity of material properties and the induced deformation gradient are responsible for the global toughening and stiffening. The discussion in Lines 206-215 is commendable, but it could be further strengthened by a more detailed quantitative theoretical model or analysis. For example, if these patterned stiffer cellular domains can be treated as inclusions in the hydrogel film, what are their volume fractions, what is the magnitude of the strain energy stored inside the cellular domains relative to that in the matrix before break, and how does the increased modulus compare to a classic theory such as Eshelby's inclusion theory?

Other comments:

1. Lines 109-110: What is the intensity and wavelength of the ultraviolet light used for polymerization?
2. Lines 118 and 324: I suggest explicitly defining the shrinking ratio A_{45C}/A_0 in the method section rather than only in the caption of Fig. S1, considering the importance of this material parameter in tuning the properties of the hydrogel film.
3. Lines 364-378: What are the operating temperature and conditions used to produce the conversion of 2D films into prescribed 3D structures? Does the out-of-plane buckling occur spontaneously at the responsive temperature (45 degree C)? Some additional mechanical details in the methods or the caption of Figure 4 would be helpful.
4. Lines 137-144 and Figure 1 d-g: The real-time FFT analysis of the dynamic responses of cellular domains in hydrogel films is intriguing. However, the change of the characteristic length scales L_a and L_b (before break) in the reciprocal space seems to be subtle compared to the undeformed sample. I am curious to see how the Poisson ratio estimated using L_a and L_b from FFT quantitatively compares to an estimation from DIC or the theoretical prediction in Supplementary Figure 2 for different subdomains.
5. Supplementary Figure 4: Could the author explain the cause of the discontinuity in some stress-strain constitutive curves? Have the authors observed any multistable phase transition in their patterned cellular structures, like those reported in metamaterials with lattice-like structures?
6. Supplementary Figure 9: Were these Poisson's ratios determined from the DIC strain visualization? Please add clarification in the caption.
7. Figure 4: It took me a long time to comprehend this figure. I find part (d) to be particularly confusing. The materials are of a uniform thickness, so why are the systems shown with a height gradient?

Minor points:

1. Figure 1: Some of the labels are hard to discern from the background. Please use different colors or labels to improve the contrast.
2. Figure 3: Please clarify in the caption what the labels P1-P5 in a-e are referring to.
3. Line 156: "representing by" should be "represented by".
4. Figures 2-4 are generally very hard to read – very small fonts and multiple different themes addressed in a single figure. So perhaps that's a style choice, but attention to readability, font sizes ... would make these results more accessible.

- What are the noteworthy results?

The combination of stiffer cell walls with softer interior results in materials with enhanced toughness relative to just the pure stiffer or softer materials. The overall approach lends itself to multiple combinations (many of which are explored in this paper) which can be tailored to attain different properties, for example positive vs. negative average Poisson's ratios. The idea of varying the cell spacing to attain differential thermal response leading to domes, saddles and other shapes is very clever – one can envision lots of variations and potential applications.

- Will the work be of significance to the field and related fields? How does it compare to the established literature? If the work is not original, please provide relevant references.

I believe that this work is significant and will inspire a lot of experimentation on this and related approaches. As mentioned above there is already a lot of work on 3D printed meta-materials, but I am not aware of any other paper in which this approach is taken.

- Does the work support the conclusions and claims, or is additional evidence needed?

The conclusions are pretty general, but they are qualitative supported. There is little real analysis that would help to understand how the systems work or from which one could glean design rules.

- Is the methodology sound? Does the work meet the expected standards in your field?

Yes

- Is there enough detail provided in the methods for the work to be reproduced?

Generally, yes, but the FEM analysis section lacks enough detail (i.e. a table of mechanical properties) to be able to reproduce the work. No details of the structures in figure 4 are provided – certainly nowhere nearly enough to reproduce the work.

Reviewer #3

(Remarks to the Author)

Reviewer #4

(Remarks to the Author)

In this study, the authors have developed a smart hydrogel system with strain-engineered heterogeneous subdomains, which enables the concurrent customization and decoupling of mechanical and deformable behaviors. This new material system shows potential for certain applications such as soft robotics and artificial muscles. Below are some comments and questions that need clarification:

1. The use of a dual crosslinker system is shown to enable a broad range of shrinking ratios. The authors demonstrate that the shrinking ratio increases with UV exposure time, as shown in Fig. S1. It would be worthwhile to extend the exposure time further to see if the shrinking ratio reaches a peak or eventually stabilizes.
2. By designing different cellular patterned domains (HC-160-L and HC-131-L), the authors claim that the material exhibits pronounced anisotropic mechanical properties. It is recommended that the authors compare their results with existing research to illustrate how significant this anisotropy is.
3. On pages 7-8, the authors note that extending the UV curing time in the film domain from 20 to 40 seconds results in increased ultimate strength, but decreased material toughness compared to the non-patterned counterpart. They attribute this to the smaller disparity in elastic modulus between stiffer patterned domains and softer film domains. However, the underlying physical mechanisms are not clearly explained. Why does increasing the UV curing time decrease the modulus difference between stiffer and softer film domains, and how does this smaller difference affect toughness?
4. On page 14, the authors claim that “with smaller subdomains and more abundant interfacial boundaries, the enhanced interface effect more effectively mediated the contrasting local characteristics, resulting in more robust hydrogels.” Are there any limitations to the size of these small subdomains? If the subdomains reach a very small scale, does this mechanism still hold?
5. In Fig. 4g, the authors need to explain why the force-displacement data does not start from zero.
6. In the study, the authors varied the UV curing time from 20s to 200s, but there is no mention of the power of the UV light; it will be difficult for interested readers to replicate the experiment.

Reviewer #5

(Remarks to the Author)

Version 1:

Reviewer comments:

Reviewer #1

(Remarks to the Author)

My concerns have been well addressed. It is a nice work. An acceptance of this manuscript for publication in this journal is recommended.

Reviewer #2

(Remarks to the Author)

I still think that the title does not reflect what is actually in the paper. Strain does not induce anything in this system. There are heterogeneous stiffness subdomains that will cause heterogeneity of the strain. The cause and effect are, in my reading, backwards in the title

The Abstract language is a bit hyperbolic. “intricate mechanical interactions: could just be “mechanical interactions.” Similarly “sophisticated mechanical behaviors” could just be “mechanical behaviors”

I have no idea what the excerpt below means:

“... we introduce a subdomain-interface mechanism that allows for the concurrent customization and decoupling of mechanical and deformable behaviors within a single material system ...”

What is the difference between “mechanical” and “deformable”? Please try to say what you mean.

The figures are improved and now easier to read – thanks for that!

Reviewer #3

(Remarks to the Author)

Reviewer #4

(Remarks to the Author)

The authors have sufficiently addressed our previous comments. The manuscript is recommended to be accepted.

Reviewer #5

(Remarks to the Author)

Point-to-Point Response to Reviewers

Reviewer #1 (Remarks to the Author):

In this manuscript, the authors developed a kind of smart hydrogels by forming strain-engineered heterogeneous subdomains within a single material system. In-situ imaging characterization was used to explore the multiscale heterogeneities and explain the intricate behaviors. It is very impressive to see both negative and positive x-direction strains. The contents are well organized. Although this work is of good innovation, there are some problems to be addressed by appropriate revisions. I recommend an acceptance of this manuscript after minor revisions. Other comments and issues are listed below.

Response: We sincerely thank the reviewer for carefully reading our manuscript with highly constructive comments and supporting the publication in *Nature Communications*. We appreciate the insightful questions, which have motivated us to further strengthen our manuscript.

Comment 1. In the introduction part, the authors mentioned the complexity of biological tissues and the crystallographic features of inorganic metals, but the gels prepared in this work are structurally mimicking the metals. The relationship between the gels and the biological tissues needs to be clarified.

Response: We thank the reviewer for the comment. To enhance readability and maintain focus on the core content, we have removed the discussions related to the analogy with metal polycrystalline structures and their strengthening mechanisms in the result sections. Instead, we now emphasize our synthetic smart hydrogel material system featuring designed stiff cellular patterns (*e.g.*, cellular domains) that induce strain-driven heterogeneous subdomains within the soft film domains under applied mechanical force. These strain-engineered heterogeneous subdomains function as unit building blocks, akin to the roles of integrated cellular structures in living organisms or cellular grain domains in hard polycrystalline materials, and are crucial in determining both the local and global properties. Consequently, we have revised the abstract, introduction and conclusion to align with the main focus of this manuscript.

Comment 2. It is very good to use in-situ imaging to characterize the structures under loadings. The conclusions obtained from this characterization should be explained more clearly. In addition, more detailed experiment procedures should be described.

Response: We thank reviewer for pointing this out. More detailed discussions regarding the in-situ imaging characterizations were added to the revised manuscript and supplementary materials. This includes: (1) analysis results (*e.g.*, Table. R1, Supplementary Table 1) from the time-resolved FFT images to demonstrate the distinct dynamic deformations; (2) calculations of strain energy density based on the full-field strain mapping to explain the enhancement of global mechanical properties (Supplementary Text). The conclusion has also been revised accordingly.

Additionally, we have provided more detailed experiment procedures in the Materials and Methods. For example: In-situ DIC analyses were conducted using GOM Correlate software (Trilion, PA, USA). A 12 MP 2D sensor with a minimum spatial resolution of 25 μm (refers to the minimum area over which the strain can be calculated) was positioned in front of the hydrogel samples to capture full-field surface images during tensile tests for DIC analysis. The image of the undeformed sample served as a reference. To better evaluate heterogeneous mechanical responses via strain mappings, Amine Functionalized Graphene Oxide (A-GO $\sim 15 \mu\text{m}$) powders, acting as markers, were mixed with the resin (0.6 mg mL^{-1}): the measured marker size ($\sim 20 \mu\text{m}$) < point distance between facets ($\sim 30 \mu\text{m}$) < measured minimum structural dimensions (~ 100

μm). The DIC capture frequency was set at 2 Hz. Real-time, full-field displacements and strains were derived from time-resolved 2D grayscale images, enabling detailed dynamic mechanical analysis at specific sites. The full-field strain mapping results were presented in two ways: (1) as superimposed images combining the optical grayscale images of the hydrogel samples with strain mapping to simultaneously show both structural and deformation information (*e.g.*, Figs. 1c, 3g, 4b, 5b, Supplementary Figs. 5, 8, 11, 14, 16, 18), or (2) as colorful strain mapping images without overlapping the optical image of the sample to better visualize the heterogeneous subdomains (*e.g.*, Fig. 2d, e, h, i, and Fig. 3a, c, e).

Table R1. Variations in basic parameters of hydrogel samples in reciprocal space during stretching.

	L'_a/L_a	L'_b/L_b	Poisson's ratio
HC-50	0.826	0.940	-
HC-90	0.767	1.133	+
HC-131	0.814	1.104	+
HC-160	0.870	1.015	+

Notes: L_a and L_b represent the initial parameters of the hydrogel samples in the undeformed state, while L'_a and L'_b represent the parameters of the hydrogels in the deformed state before failure.

Comment 3. The gel materials are composed of hard skeleton and soft domains. It seems that the main properties of this kind of materials are contributed by the hard skeleton. What is the role of soft domains? How about the properties of the structure with only the hard skeleton?

Response: We sincerely thank the reviewer for this comment. We agree with the reviewer that the stiff cellular patterns contribute to the enhanced mechanical properties (*e.g.*, increased elastic modulus) of the overall hydrogel samples. Unlike previous studies that only rely on incorporating reinforced inclusions into soft matrices to enhance material properties, the stiff pattern domain in our hydrogel materials can induce strain-driven heterogeneous subdomains within the hydrogel film (*e.g.*, soft film domain), resulting from the distinct mechanical behaviors between the pattern and film domains under applied mechanical force. We have demonstrated that the interplay among the localization, integration, and interaction of these subdomain building blocks within the soft film allows for extensive tuning of both local and global behaviors.

Furthermore, we also added the calculations of the strain energy density across different local regions, including subdomains A and B within the ductile film domain and the stiff pattern domains (see details in Supplementary Text). We demonstrated that these heterogeneous subdomains endured greater local deformations and absorbed more strain energy than that of the homogeneous thin film counterpart cured under the same photo-polymerization dose, leading to enhanced global material toughness in our patterned hydrogels. Beyond the reinforcement provided by the stiff pattern domains, the greater strain energy stored in subdomains B and their large biaxial local strain, due to the negative Poisson's ratio effect of the meta-structured patterns, offer enhanced resistance to deformation and stress. This contributes to an improvement in the global elastic modulus and ultimate strength (ref. 33).

In our upcoming work, we have thoroughly examined the mechanical properties and shape transformation of metamaterial frameworks (without film), which will be published soon. Generally, these frameworks exhibit a significantly lower elastic modulus compared to our patterned hydrogel films, and their strain response and shape morphing behavior differ from the hydrogels presented in this study.

Comment 4. In page 5, the authors said that PEGDA will crosslink the polymer first and then the BIS. Why the crosslinking does not happen at the same time?

Response: We sincerely thank the reviewer for this comment. The PEGDA and BIS crosslink the polymer at the same time, but at different reaction rates. We refer to the propagation reaction rate in Ref. 27, $R_p = k_p[M][M_n \bullet]$, where k_p represents the propagation rate constant, $[M]$ represents the double bond concentration, and $[M_n \bullet]$ represents the total propagating radical concentration. Considering PEGDA and BIS both contain two double bonds at their ends and the concentration of PEGDA is higher than that of BIS, $[M_{C=C \text{ from PEGDA}}] > [M_{C=C \text{ from BIS}}]$, thus PEGDA dominates the earlier polymerization.

Thus, we revised the associated discussion in the manuscript as following “Specifically, the quick gel formation, primarily due to long-chain PEGDA crosslinking of the major monomer NIPAm, allowed for the creation of a hydrogel film within a short exposure period (refs. 26, 27). As the curing time increased, the polymer network continued growing, involving more minor monomer AAm and short-chain BIS, leading to copolymer hydrogel films crosslinked with both PEGDA and BIS that showed enhanced ultimate strength and elastic modulus with longer UV exposure times (Supplementary Fig. 1a) (ref. 28).”

Comment 5. In Figure 3j, the authors claimed that ultimate strength and the inverse square root of the critical dimensions showed linear relationship. Is there any physical meaning of this scale relationship? For better visualization, a semi-log plot should be more appropriate.

Response: We sincerely thank the reviewer for this comment. These linear correlations are similar to the Hall-Petch relationship observed in polycrystalline materials (ref. 8), they uncover different underlying mechanisms. In our material system, we have demonstrated that the interplay of localization and interaction between stiff pattern-driven heterogeneous subdomains as well as their interfaces influences both local and global properties. Here, we further proved a subdomain-interface mechanism that “with smaller subdomains and more abundant interfacial boundaries (Supplementary Fig. 16), the enhanced interface effect more effectively mediated the contrasting local characteristics, thereby manipulating the global mechanical properties.” We replotted the results in semi-log forms in the Fig. 3j and Supplementary Fig. 17, which are also shown below in Fig. R1 and Fig. R2.

Fig. R1. Ultimate strength versus critical dimensions (e.g., d , d_A , and d_B) across a series of HC-90 (a) and HC-50 hydrogel films (b) with varying dimensions.

Reviewer #2 (Remarks to the Author):

This paper is based on the clever idea patterning a hydrogel into a cellular pattern of stiffer and less stiff regions. It is somewhat analogous to the extensive work on 3D printed meta-materials except that the effect results from differential crosslinking and instead of the open space in 3D printed materials the spaces between the stiff cell walls are filled with a lower modulus material. The resulting materials exhibit customizable mechanical properties, enhanced strength and toughness, and thermo-responsive deformations.

The authors return several times to an analogy to metal polycrystalline materials. This analogy, in my opinion, is overstated and not necessary to the point of the paper. For example the word “precipitate” is used though there are in fact zero precipitates in the systems studied. There is an analogy to the Hall-Petch relation that has no physical basis. I would recommend eliminating this analogy – it would make the paper easier to read and not distract from the really interesting elements of this work. Along similar lines the analogies to biological materials don’t really add much to the paper. How would the structures designed here resemble actual hierarchical structure of biomaterials?

On the title of the paper: I don’t see anywhere that strain induces heterogeneous subdomains. Rather the subdomains are baked in by the patterning due to differential polymerization. In my opinion the title needs revision, perhaps to something like “Tailoring Smart Hydrogels Through Manipulation of Subdomains of Local Stiffness.”

Response: We sincerely thank the reviewer for carefully reading our manuscript and providing constructive comments and suggestions. We welcome the opportunities to address these questions and describe the changes we have made accordingly.

We agree with the reviewer that this analogy between our designed hydrogel system and the polycrystalline materials was overstated. To make this paper easier to read and not distract from the essential content, we have removed all the discussions regarding the analogy to metal polycrystalline structures and strengthening mechanisms and made corresponding revisions to emphasize the major element of this work.

Specifically, our synthetic smart hydrogel material system featuring engineered stiff cellular patterns (*e.g.*, cellular domains) can induce strain-driven heterogeneous subdomains within the soft film domains under applied mechanical force. These subdomains emerge due to the distinct elastic modulus and mechanical behaviors between the pattern and film domains upon external mechanical force. Importantly, these strain-engineered heterogeneous subdomains function as unit building blocks, akin to the roles of integrated cellular structures in living organisms or cellular grain domains in hard polycrystalline materials, playing an essential role in determining both local and global properties. We also emphasized the hierarchical heterogeneities in our patterned hydrogel materials, which are represented by downscale localization (*e.g.*, heterogeneous subdomains) and upscale integration of various cellular patterns (*e.g.*, different “phases”), as well as molecular-level dual crosslinked networks. The spatiotemporal control of photo-polymerization, which seamlessly incorporates these hierarchical heterogeneities, allows for extensive tuning of both local and global features. Thus, the successful application of a broad spectrum of structural complexities at hierarchical scales to our synthetic hydrogels offers a new strategy for designing soft materials, diverging from the conventional molecular-level design of polymer materials. The abstract, introduction, conclusion, and associated discussions were revised to focus on these main ideas (see highlights in the revised manuscript).

Given the critical role of strain-induced subdomain building blocks in controlling both local and global features, we would like to highlight this aspect in the title.

An additional recommendation is that the authors do more to strengthen the mechanical aspects of the study, particularly related to the mechanics underlying the observed stiffening and toughening effects in the patterned hydrogel composite structures. While these effects are evident from uniaxial tension tests and anticipated based on the material properties of the subdomains compared to the matrix, it is not immediately clear how such local heterogeneity of material properties and the induced deformation gradient are responsible for the global toughening and stiffening. The discussion in Lines 206-215 is commendable, but it could be further strengthened by a more detailed quantitative theoretical model or analysis. For example, if these patterned stiffer cellular domains can be treated as inclusions in the hydrogel film, what are their volume fractions, what is the magnitude of the strain energy stored inside the cellular domains relative to that in the matrix before break, and how does the increased modulus compare to a classic theory such as Eshelby's inclusion theory?

Response: We thank the reviewer for this comment. According to reviewer's recommendation, we have calculated the strain energy density for HC-50 hydrogel films with soft film domains exposed to 20 seconds and 40 seconds. Detailed calculations have been added to the Supplementary Text, and the related discussions are now included in the revised manuscript (see the highlights), which is also briefly summarized below.

"To further investigate the impact of localized heterogeneities on global mechanical properties, we calculated the strain energy density across different local regions, including subdomains A and B within the ductile film domain and the stiff pattern domains. At a stretch of 1.2 (20% global strain), subdomains A and B in the HC-50 sample experienced larger local deformations ($\epsilon_A = 45 - 57\%$ and $\epsilon_B = 17 - 41\%$) compared to the homogeneous thin film ($\epsilon_{film-20s} = 20\%$) cured for the same duration (20 seconds). Consequently, the strain energy densities of these heterogeneous subdomains were $u_A = 6.0 \text{ J m}^{-3}$ and $u_B = 3.3 \text{ J m}^{-3}$, which are five to ten times those of the homogeneous thin film under the same global strain of 20% ($u_{film-20s} = 0.6 \text{ J m}^{-3}$). These elevated strain energy densities in the local subdomains contribute to the increased material toughness of the overall patterned hydrogels.

In addition to reinforcement from the stiff pattern domains (elastic modulus $E_{200s} = 635 \text{ kPa}$), the high strain energy density in subdomains B, combined with the biaxial strain caused by the negative Poisson's ratio effect of the meta-structured patterns, offered enhanced resistance to deformation and stress. This led to improvements in both the global elastic modulus and ultimate strength, consistent with previous studies showing that auxetic inclusions can enhance the overall composite's mechanical properties (ref. 33). The interaction and integration of these localized subdomains enable extensive tuning of both local and global mechanical behaviors.

However, increasing the UV curing time of the film domain reduced the localization effects on global mechanical properties (Supplementary Fig. 5). Specifically, as UV exposure increased in the soft film regions, the strain energy density in subdomains A of the HC-50 hydrogels decreased from 6.0 J m^{-3} (20 seconds) to 3.8 J m^{-3} (40 seconds), indicating a reduced discernibility of the strain-induced heterogeneities within the soft domains. Meanwhile, a higher strain energy density was calculated in the vertical regions of the stiff patterns (3.8 J m^{-3} for 20 seconds vs. 6.2 J m^{-3} for 40 seconds) as UV exposure increased in the soft film domains, suggesting that the stiff patterns increasingly bore the load rather than inducing localized heterogeneous strain in the soft film domains. Therefore, a substantial mismatch in elastic modulus between the pattern and film domains is crucial for inducing localized heterogeneous subdomains, which leads to a synergy between strength and toughness. These findings underscore the effectiveness of cellular pattern designs in controlling local subdomains and their collective attributes at the macroscopic level, offering a new approach to developing synthetic soft materials with tailored mechanical features."

To simplify the calculations of strain energy density (u_i), we made several assumptions that may lead to discrepancies from the actual case. For example, we used the elastic moduli of homogeneous hydrogel films (cured for 20, 40, and 200 seconds) to represent the elastic moduli in the local pattern (200 seconds) and film domains (20 or 40 seconds) when calculating the strain energy density of the patterned hydrogels. In our material system, resistance to deformation in the patterned samples may vary across different regions within the pattern or film domains, leading to variations in elastic moduli across local regions. We also estimated the strain profiles within the x-y plane (using average strain values for $u_{inclined}$ and $u_{vertical}$; or by integrating simplified strain profiles for u_A and u_B) to calculate the strain energy density in localized domains. As a result, the calculated sum of the weighted strain energy densities across local domains ($U_i = \sum_1^i u_i \cdot A_i, i = 1, 2, 3 \dots$) based on volume/area fractions and local strain energy densities was lower than the strain energy density directly derived from the stress-strain curves of the overall patterned hydrogel samples (HC-50 and HC-90).

A more comprehensive understanding of the mechanisms governing these complex structures and behaviors requires further theoretical studies and analysis. We plan to work together with our collaborators to explore this in greater depth in future work.

Other comments:

Comment 1: Lines 109-110: What is the intensity and wavelength of the ultraviolet light used for polymerization?

Response: We thank the reviewer for pointing this out. We have included the details regarding the intensity and wavelength of the ultraviolet light used in the 3D printer in the Materials and Methods section “Digital light projection grayscale lithography was performed using an Anycubic Photon D2 digital light processing (DLP)-3D printer (Intensity and wavelength of the ultraviolet light: 2.5 mW cm⁻², 405nm).”

Comment 2: Lines 118 and 324: I suggest explicitly defining the shrinking ratio A_{45C}/A_0 in the method section rather than only in the caption of Fig. S1, considering the importance of this material parameter in tuning the properties of the hydrogel film.

Response: We thank the reviewer for this comment. We have shown the details in “Materials and Methods” section: “**Measurement of areal swelling and shrinking ratios.** Homogeneous films (5 mm × 5 mm) underwent uniform UV exposure for 8-300 seconds to study the swelling and shrinking ratios of the hydrogels (Supplementary Fig. 1). Swelling measurements involved placing the printed samples in DI water at 25 °C for 6 hours until stable. The swelled state area ($A_{25°C}$) was measured, and areal swelling ratios were calculated as $A_{25°C}/A_0$, where A_0 is the area of as-printed hydrogel film. For shrinking measurements, NIPAm-AAm samples were first hydrated at 25°C for 2 hours, then gradually heated to 45 °C (5 °C per hour). A similar process was applied to NIPAm samples, heating to 35 °C (0.5 °C per hour). Surface area measurements were taken using a Trinocular Stand Stereo Zoom Microscope (7X-45X, AmScope) and analyzed by ImageJ (NIH, USA).”

Comment 3: Lines 364-378: What are the operating temperature and conditions used to produce the conversion of 2D films into prescribed 3D structures? Does the out-of-plane buckling occur spontaneously at the responsive temperature (45 degree C)? Some additional mechanical details in the

methods or the caption of Figure 4 would be helpful.

Response: We thank the reviewer for this comment. For the PNIPAm-co-AAm copolymer hydrogel, deformation begins at 35 °C and stabilizes at 45 °C, as shown in Figure. 4e. For the PNIPAm hydrogel, deformation starts at 25 °C and stabilizes around 35 °C, close to physiological temperature, as seen in Figure 4h and Supplementary Figure 22. In response to the reviewer’s comment, we have added the following description to the revised manuscript: “The resulting 3D shapes are formed through a sequential 2D-to-3D shape transformation within the temperature range of 35 °C to 45 °C.”

Comment 4: Lines 137-144 and Figure 1 d-g: The real-time FFT analysis of the dynamic responses of cellular domains in hydrogel films is intriguing. However, the change of the characteristic length scales L_a and L_b (before break) in the reciprocal space seems to be subtle compared to the undeformed sample. I am curious to see how the Poisson ratio estimated using L_a and L_b from FFT quantitatively compares to an estimation from DIC or the theoretical prediction in Supplementary Figure 2 for different subdomains.

Response: We thank the reviewer for this valuable question. While we do not obtain quantitative values for strains and Poisson’s ratios, the unique reciprocal-space patterns derived from FFT analysis provide a powerful tool for fast real-time monitoring of dynamic responses under external force. These patterns allow for easy detection of time-resolved variations in basic parameters (e.g., L'_a/L_a , L'_b/L_b), indicating whether a region is undergoing contraction or elongation, as well as positive or negative Poisson’s ratios. More importantly, FFT analysis distinguishes the distinct behaviors of different local “phases” within a single sample. For example, two sets of time-resolved FFT patterns reveal distinct responses in reciprocal space during stretching, tracking the different deformations between the stiffer “HC-160-L phase” (marked by yellow ovals) and the softer “HC-50 phase” (highlighted by pink circles) (Fig. 3b). Based on dynamic FFT analysis, the variations in basic parameters for the four different cellular designs are listed below:

Table R1. Variations in basic parameters of hydrogel samples in reciprocal space during stretching.

	L'_a/L_a	L'_b/L_b	Poisson’s ratio
HC-50	0.826	0.940	-
HC-90	0.767	1.133	+
HC-131	0.814	1.104	+
HC-160	0.870	1.015	+

Notes: L_a and L_b represent the initial parameters of the hydrogel samples in the undeformed state, while L'_a and L'_b represent the parameters of the hydrogels in the deformed state before failure.

Comment 5: Supplementary Figure 4: Could the author explain the cause of the discontinuity in some stress-strain constitutive curves? Have the authors observed any multistable phase transition in their patterned cellular structures, like those reported in metamaterials with lattice-like structures?

Response: We appreciate the reviewer’s comment. To clarify, all stress-strain curves are represented by solid lines. The discontinuities observed in some of the curves are caused by the onset of fracture. We did not observe multi-stable phase transitions in our patterned hydrogel films. While the engineered cellular patterns resemble typical metamaterials, the strong interactions between the stiff cellular domain and the soft film domain likely limit the possibility of multi-stable phase transitions.

Comment 6: Supplementary Figure 9: Were these Poisson's ratios determined from the DIC strain visualization? Please add clarification in the caption.

Response: We thank the reviewer for this comment. Yes, the Poisson's ratios were determined from the DIC strain visualization. We have included the details in "**Materials and Methods**" section: "The local strains (ε_x and ε_y) and their time-dependent variations for any specific sites were derived from time-resolved full-field strain maps. Poisson's ratios of subdomains were calculated using the formula $\nu = -\frac{d\varepsilon_x}{d\varepsilon_y}$. This involved selecting four key tracking points at the left, right, bottom, and top. These points allowed us to measure deformations along the x and y axes, calculated as $\Delta x = x_{right} - x_{left}$ and $\Delta y = y_{top} - y_{bottom}$, respectively (Supplementary Fig. 9)."

We added a description to the caption of Supplementary Fig. 9: "The Poisson's ratios were determined from the full-field DIC strain maps."

Comment 7: Figure 4: It took me a long time to comprehend this figure. I find part (d) to be particularly confusing. The materials are of a uniform thickness, so why are the systems shown with a height gradient?

Response: We thank the reviewer for this comment. We clarified in the caption of Fig. 4d: "Designs of non-Euclidean caps incorporating various arrangements of heterogeneous "phases": spatial distribution of curing times (d) and resulting 3D caps after shape transformation (e)." The varying heights in the 3D models (Fig. 4d) correspond to differences in accumulated UV curing times, which directly influence the shrinking ratio distribution. This can also be represented in a 2D format, with colors denoting curing times (e.g., t_{ex} distribution in Supplementary Fig. 21). To ensure clarity, we applied color to the 3D models to highlight the spatial distribution of accumulated UV curing times.

Minor points:

Comment 1: Figure 1: Some of the labels are hard to discern from the background. Please use different colors or labels to improve the contrast.

Response: Thank you for the suggestion. We have improved the contrast in all figures to make the labels easier to read.

Comment 2: Figure 3: Please clarify in the caption what the labels P1-P5 in a-e are referring to.

Response: We thank the reviewer for this comment. We have added descriptions of P1-P6 in the caption of Figure 3. For example, P1, P3, and P4 refer to subdomain A of the "HC-50 phase", P2 refers to the film domain of the "HC-160-L phase", P5 and P6 refer to the central region and the area near the interface within the "HC-160-T inclusion phase".

Comment 3: Line 156: “representing by” should be “represented by”.

Response: We thank the reviewer for this comment. We have corrected it.

Comment 4: Figures 2-4 are generally very hard to read – very small fonts and multiple different themes addressed in a single figure. So perhaps that’s a style choice, but attention to readability, font sizes ... would make these results more accessible.

Response: We thank the reviewer for pointing this out. We have improved the qualities of all figures including the colors, font sizes, and others.

- What are the noteworthy results?

The combination of stiffer cell walls with softer interior results in materials with enhanced toughness relative to just the pure stiffer or softer materials. The overall approach lends itself to multiple combinations (many of which are explored in this paper) which can be tailored to attain different properties, for example positive vs. negative average Poisson’s ratios.

The idea of varying the cell spacing to attain differential thermal response leading to domes, saddles and other shapes is very clever – one can envision lots of variations and potential applications.

- Will the work be of significance to the field and related fields? How does it compare to the established literature? If the work is not original, please provide relevant references.

I believe that this work is significant and will inspire a lot of experimentation on this and related approaches. As mentioned above there is already a lot of work on 3D printed meta-materials, but I am not aware of any other paper in which this approach is taken.

- Does the work support the conclusions and claims, or is additional evidence needed?

The conclusions are pretty general, but they are qualitative supported. There is little real analysis that would help to understand how the systems work or from which one could glean design rules.

- Is the methodology sound? Does the work meet the expected standards in your field?

Yes

- Is there enough detail provided in the methods for the work to be reproduced?

Generally, yes, but the FEM analysis section lacks enough detail (i.e. a table of mechanical properties) to be able to reproduce the work. No details of the structures in figure 4 are provided – certainly nowhere nearly enough to reproduce the work.

Response: We thank the reviewer for this comment. According to the reviewer’s comment, we have added “Supplementary Table 3. Key parameters used in the finite element method (FEM) simulation,” which is shown below in Table R2. We have also used colored 3D models to represent the spatial distribution of accumulated exposure time in designed 3D caps in Figure 4, which is attached below in Fig. R2.

Table R2. Key parameters used in the finite element analysis (FEA) simulation.

Sample for FEM simulation	Parameter	Region	Value
HC-50 within 20s film	Elastic modulus (kPa)	Pattern domain	635
		Film domain	35
	L (mm)		2.211
	W (mm)		1.105
	t (mm)		0.32
HC-90 within 20s film	Elastic modulus (kPa)	Pattern domain	635
		Film domain	35
	L (mm)		2.25
	W (mm)		0.975
	t (mm)		0.38
HC-50 within 40s film	Elastic modulus (kPa)	Pattern domain	635
		Film domain	180
	L (mm)		2.8
	W (mm)		1.4
	t (mm)		0.35
HC-90 within 40s film	Elastic modulus (kPa)	Pattern domain	635
		Film domain	180
	L (mm)		2.3
	W (mm)		1
	t (mm)		0.38
HC-120 within 20s film	Elastic modulus (kPa)	Pattern domain	635
		Film domain	180
	L (mm)		2.55
	W (mm)		1.27
	t (mm)		0.31
HC-160-L within 20s film	Elastic modulus (kPa)	Pattern domain	635
		Film domain	180
	L (mm)		2.7
	W (mm)		1
	t (mm)		0.2

Notes: the length of the vertical wall L , the length of the inclined wall W , the wall thickness t , and the inclined angle θ are demonstrated in Supplementary Fig. 2.

Fig. R2. Spatial distribution of curing times for non-Euclidean caps in Figure. 4e. Embedding heterogeneous phases in circle hydrogel plates with exposure time of 20 seconds, encode spherical caps using the growth function $\eta = A_{45^\circ\text{C}}/A_0 = \frac{c}{(1+R_b^2)^z}$, where c is constant, $R_b = b/R$ is the relative radius of each concentric circle, and b represents the variable radius of the circle.

Reviewer #3 (Remarks to the Author):

Comment 1: I co-reviewed this manuscript with one of the reviewers who provided the listed reports. This is part of the Nature Communications initiative to facilitate training in peer review and to provide appropriate recognition for Early Career Researchers who co-review manuscripts.

Response: We sincerely thank the reviewer for carefully reading our manuscript with highly constructive comments.

Reviewer #4 (Remarks to the Author):

Comment 1: In this study, the authors have developed a smart hydrogel system with strain-engineered heterogeneous subdomains, which enables the concurrent customization and decoupling of mechanical and deformable behaviors. This new material system shows potential for certain applications such as soft robotics and artificial muscles. Below are some comments and questions that need clarification.

Response: We thank the reviewer for carefully reading our manuscript and raising very valuable questions, which have motivated us to further improve the quality of this work.

Comment 2: The use of a dual crosslinker system is shown to enable a broad range of shrinking ratios. The authors demonstrate that the shrinking ratio increases with UV exposure time, as shown in Fig. S1. It would be worthwhile to extend the exposure time further to see if the shrinking ratio reaches a peak or eventually stabilizes.

Response: We appreciate the reviewer's comment and agree that the current shrinking ratio has not reached its maximum. However, with a current maximum value ($A_{45^\circ\text{C}}/A_0$) of 0.92, which is very close to 1.0, we believe there is limited room for further increase. The range of 0.28-0.92 is also sufficient for designing the 2D-to-3D shape transformations. Additionally, during the fabrication of the cellular pattern domains, we observed that longer exposure times caused light scattering, which compromised structural precision. Therefore, we limited the exposure time for the cellular region to 200 seconds and investigated shrinkage for exposure time up to 300 seconds.

Comment 3: By designing different cellular patterned domains (HC-160-L and HC-131-L), the authors claim that the material exhibits pronounced anisotropic mechanical properties. It is recommended that the

authors compare their results with existing research to illustrate how significant this anisotropy is.

Response: We thank the reviewer for this comment. The anisotropy in our patterned hydrogel materials is achieved through the engineering of cellular domains, which exhibit more pronounced anisotropic behavior compared to homogeneous hydrogel films without patterning. While traditional metamaterial frameworks often show stronger anisotropic characteristics due to a higher degree of freedom for large deformations, they are less effective at balancing strength and toughness compared to our patterned hydrogels, which benefit from strain-induced subdomains.

Comment 4: On pages 7-8, the authors note that extending the UV curing time in the film domain from 20 to 40 seconds results in increased ultimate strength, but decreased material toughness compared to the non-patterned counterpart. They attribute this to the smaller disparity in elastic modulus between stiffer patterned domains and softer film domains. However, the underlying physical mechanisms are not clearly explained. Why does increasing the UV curing time decrease the modulus difference between stiffer and softer film domains, and how does this smaller difference affect toughness?

Response: We thank the reviewer for this comment. In our patterned hydrogel system, we print relatively stiff cellular domains (*e.g.*, 200-second exposure) that induce strain-driven heterogeneous subdomains within the soft film domains (*e.g.*, exposed for 20 or 40 seconds). These subdomains arise from the differences in elastic modulus and mechanical behavior between the pattern and film domains when subjected to mechanical forces. Specifically, we calculated the strain energy density across different local regions, including subdomains A and B within the ductile film domain and the stiff pattern domains (see details in Supplementary Text). We demonstrated that these heterogeneous subdomains endured greater local deformations compared to the homogeneous thin film cured for the same duration (*e.g.*, 20 seconds). Consequently, the strain energy densities of these heterogeneous subdomains were $u_A = 6.0 \text{ J m}^{-3}$ and $u_B = 3.3 \text{ J m}^{-3}$, which are five to ten times those of the homogeneous thin film under the same global strain of 20% ($u_{film-20s} = 0.6 \text{ J m}^{-3}$). These elevated strain energy densities in the local subdomains contribute to the increased material toughness of the overall patterned hydrogels.

In addition to reinforcement from the stiff pattern domains (elastic modulus $E_{200s} = 635 \text{ kPa}$), the high strain energy density in subdomains B, combined with the biaxial strain caused by the negative Poisson's ratio effect of the meta-structured patterns, offered enhanced resistance to deformation and stress. This led to improvements in both the global elastic modulus and ultimate strength, consistent with previous studies showing that auxetic inclusions can enhance the overall composite's mechanical properties (ref. 33). The interaction and integration of these localized subdomains enable extensive tuning of both local and global mechanical behaviors.

However, increasing the UV curing time of the film domain reduced the localization effects on global mechanical properties (Supplementary Fig. 5). Specifically, as UV exposure increased in the soft film regions, the strain energy density in subdomains A of the HC-50 hydrogels decreased from 6.0 J m^{-3} (20 seconds) to 3.8 J m^{-3} (40 seconds), indicating a reduced discernibility of the strain-induced heterogeneities within the soft domains. Meanwhile, a higher strain energy density was calculated in the vertical regions of the stiff patterns (3.8 J m^{-3} for 20 seconds vs. 6.2 J m^{-3} for 40 seconds) as UV exposure increased in the soft film domains, suggesting that the stiff patterns increasingly bore the load rather than inducing localized heterogeneous strain in the soft film domains.

Therefore, a substantial mismatch in elastic modulus between the pattern and film domains is crucial for inducing localized heterogeneous subdomains, which leads to a synergy between strength and toughness.

Comment 5: On page 14, the authors claim that “with smaller subdomains and more abundant interfacial boundaries, the enhanced interface effect more effectively mediated the contrasting local characteristics, resulting in more robust hydrogels.” Are there any limitations to the size of these small subdomains? If the subdomains reach a very small scale, does this mechanism still hold?

Response: We thank the reviewer for this insightful comment. Our hydrogel films were printed using a commercial 3D printer, with dimensions limited by the printer’s resolution (*e.g.*, resolution = 51 μm). Additionally, light scattering further affects the printing precision. For example, the smallest designed unit cell with a clean interface has a wall thickness of 153 μm . We believe our mechanism would remain effective at smaller scales with improved printing resolution. Using high-resolution printing techniques such as micro-stereolithography or two-photon lithography could reduce the feature size to the micron or even nanoscale level.

Comment 6: In Fig. 4g, the authors need to explain why the force-displacement data does not start from zero.

Response: We thank the reviewer for this comment. The force-displacement curves for the caps submerged in water were generated by subtracting the buoyancy data from a compression test conducted without a sample. The non-zero starting point of the force data is due to slight vibrations in the buoyancy at the beginning. We included this note in the “**Materials and Methods, Mechanical tests of synthetic hydrogels**”.

Comment 7: In the study, the authors varied the UV curing time from 20s to 200s, but there is no mention of the power of the UV light; it will be difficult for interested readers to replicate the experiment.

Response: We thank the reviewer for pointing this out. We thank the reviewer for pointing this out. We have added the details of the intensity and wavelength of the ultraviolet light used in the 3D printer to the Materials and Methods section “Digital light projection grayscale lithography was performed using an Anycubic Photon D2 digital light processing (DLP)-3D printer (Intensity, and wavelength of the ultraviolet light: 2.5 mW cm^{-2} , 405nm).”

Reviewer #5 (Remarks to the Author):

Comment 1: I co-reviewed this manuscript with one of the reviewers who provided the listed reports. This is part of the Nature Communications initiative to facilitate training in peer review and to provide appropriate recognition for Early Career Researchers who co-review manuscripts.

Response: We sincerely thank the reviewer for carefully reading our manuscript with highly constructive comments, which have motivated us to further improve this manuscript.

Point-to-Point Response to Reviewers

Reviewer #1 (Remarks to the Author):

My concerns have been well addressed. It is a nice work. An acceptance of this manuscript for publication in this journal is recommended.

Response: We sincerely thank the reviewer for all the comments and suggestions. We greatly appreciate your recommendation for publication.

Reviewer #2 (Remarks to the Author):

I still think that the title does not reflect what is actually in the paper. Strain does not induce anything in this system. There are heterogeneous stiffness subdomains that will cause heterogeneity of the strain.

The cause and effect are, in my reading, backwards in the title

The Abstract language is a bit hyperbolic. “intricate mechanical interactions: could just be “mechanical interactions.” Similarly “sophisticated mechanical behaviors” could just be “mechanical behaviors”

I have no idea what the excerpt below means:

“ ... we introduce a subdomain-interface mechanism that allows for the concurrent customization and decoupling of mechanical and deformable behaviors within a single material system ...”

What is the difference between “mechanical” and “deformable”? Please try to say what you mean.

The figures are improved and now easier to read – thanks for that!

Response: We sincerely thank the reviewer for the further comments. We fully agree with the reviewer’s statement that “There are heterogeneous stiffness subdomains that will cause heterogeneity of the strain.” There may be some confusion regarding the naming of the structures. Here are the clarifications of the structures mentioned in this manuscript. The designed hydrogel films consist of relatively stiff cellular patterns (referred to as the “cellular pattern domain”) and a relatively soft film (referred to as the “film domain”). The difference in elastic modulus between the pattern domain and film domain leads to the emergence of heterogeneous subdomains within the film domain, which we have labeled as subdomains A and B when external mechanical force is applied. To improve clarity, we have revised the title to “Tailoring Smart Hydrogels Through Manipulation of Heterogeneous Subdomains” to better reflect the content of the paper. Additionally, we have removed hyperbolic words such as “intricate” and “sophisticated” in the revised manuscript.

“ ... we introduce a subdomain-interface mechanism that allows for the concurrent customization and decoupling of mechanical and deformable behaviors within a single material system ...”

We intended to emphasize the manipulation of mechanical properties and shape transformations. Specifically, in Figure 4 (panels a and c) of our manuscript (also attached below in Figure R1, a and b), our 3D model unravels the interplay among shrinkage, elastic modulus, and cellular unit sizes, providing a possible solution for simultaneously directing both shrinkage-induced deformations and mechanical properties within a single hydrogel material system. We have now refined this statement to specifically address “mechanical properties and shape transformations”.

Figure R1. **a**, Correlations between elastic moduli and shrinkage ratios for HC-90 hydrogels with varying cellular unit sizes ($d = 0.99 - 2.44$ mm) under different local curing times of cellular patterns ranging from 40 to 200 seconds. **b**, A 3D master surface displaying the correlations among shrinkage ratios, elastic moduli, and cellular unit sizes.

Reviewer #3 (Remarks to the Author):

Response: We sincerely appreciate reviewer's previous comments and recommendation for publication.

Reviewer #4 (Remarks to the Author):

The authors have sufficiently addressed our previous comments. The manuscript is recommended to be accepted.

Response: We sincerely thank the reviewer for constructive comments and recommendation for publication.

Reviewer #5 (Remarks to the Author):

Response: We sincerely thank the reviewer for all the comments and suggestions. We greatly appreciate your recommendation for publication.